# Phenotype integration improves power and preserves specificity in biobank-based genetic studies of major depressive disorder

Andrew Dahl [1,15] ✉, Michael Thompson[2], Ulzee An [2], Morten Krebs [3], Vivek Appadurai [3], Richard Border [2,4,5], Silviu-Alin Bacanu [6], Thomas Werge [3,7,8], Jonathan Flint [4], Andrew J. Schork[3,9,10], Sriram Sankararaman [2,4,11], Kenneth S. Kendler [6] & Na Cai[12,13,14,15] ✉

Biobanks often contain several phenotypes relevant to diseases such as major depressive disorder (MDD), with partly distinct genetic architectures. Researchers face complex tradeoffs between shallow (large sample size, low specificity/sensitivity) and deep (small sample size, high specificity/ sensitivity) phenotypes, and the optimal choices are often unclear. Here we propose to integrate these phenotypes to combine the benefits of each. We use phenotype imputation to integrate information across hundreds of MDD-relevant phenotypes, which significantly increases genome-wide association study (GWAS) power and polygenic risk score (PRS) prediction accuracy of the deepest available MDD phenotype in UK Biobank, LifetimeMDD. We demonstrate that imputation preserves specificity in its genetic architecture using a novel PRS-based pleiotropy metric. We further find that integration via summary statistics also enhances GWAS power and PRS predictions, but can introduce nonspecific genetic effects depending on input. Our work provides a simple and scalable approach to improve genetic studies in large biobanks by integrating shallow and deep phenotypes.

Although major depressive disorder (MDD) is the most common psychiatric disorder and the leading cause of disability worldwide, its causes are largely unknown[1,2]. Despite the moderate familial heritability of MDD (~40%)[3], genome-wide association studies (GWASs) have only recently begun to identify replicable risk loci and polygenic risk scores (PRSs)[4–9]. These discoveries were enabled by increasing power along two primary dimensions: depth of phenotyping and sample size[1]. Increasing sample size improves both GWAS and PRS power by

[1]Section of Genetic Medicine, University of Chicago, Chicago, IL, USA. [2]Department of Computer Science, University of California, Los Angeles, Los Angeles, CA, USA. [3]Institute of Biological Psychiatry, Mental Health Center–Sct Hans, Copenhagen University Hospital–Mental Health Services CPH, Copenhagen, Denmark. [4]Department of Human Genetics, David Geffen School of Medicine, University of California, Los Angeles, Los Angeles, CA, USA. [5]Department of Epidemiology, Harvard T.H. Chan School of Public Health, Boston, MA, USA. [6]Virginia Institute for Psychiatric and Behavioral Genetics and Department of Psychiatry, Virginia Commonwealth University, Richmond, VA, USA. [7]Lundbeck Foundation GeoGenetics Centre, Natural History Museum of Denmark, University of Copenhagen, Copenhagen, Denmark. [8]Department of Clinical Medicine, Faculty of Health and Medical Sciences, University of Copenhagen, Copenhagen, Denmark. [9]Neurogenomics Division, The Translational Genomics Research Institute (TGEN), Phoenix, AZ, USA. [10]Section for Geogenetics, GLOBE Institute, Faculty of Health and Medical Sciences, Copenhagen University, Copenhagen, Denmark. [11]Department of Computational Medicine, University of California, Los Angeles, Los Angeles, CA, USA. [12]Helmholtz Pioneer Campus, Helmholtz Zentrum München, Neuherberg, Germany. [13]Computational Health Centre, Helmholtz Zentrum München, Neuherberg, Germany. [14]School of Medicine, Technical University of Munich, Munich, Germany. [15]These authors contributed equally: Andrew Dahl, Na Cai. ✉e-mail: andywdahl@uchicago.edu; na.cai@helmholtz-munich.de

reducing the standard errors of estimated genetic effects on a given MDD phenotype[4,10]. Alternatively, increasing diagnostic accuracy through structured clinical interviews prevents dilution of genetic effect sizes, thus improving GWAS power[1,9,11] and PRS accuracy[11,12]. In practice, studies have a fixed budget and must trade off between increasing sample size or phenotyping depth. The optimal choice for current and future MDD studies remains contested[11,13,14]. Ultimately, the choice will depend on the study's goals.

One important goal is statistical explanation, defined as the number of GWAS hits or the PRS prediction accuracy. Most MDD GWASs have focused on this goal, which is best achieved by maximizing sample size[11,12]. This motivates the use of shallow phenotypes in large biobanks, including self-reported depression and treatment[5,7]. Sample sizes are often further increased by including health record information of seeking care for depression (for example, the Integrative Psychiatric research consortium (iPSYCH[15]) and the Million Veterans Program[8]). These studies have amassed sample sizes of millions of individuals and have identified hundreds of risk loci, as well as PRSs with state-of-the-art prediction accuracy in European-ancestry clinical cohorts[4–8].

A partly distinct goal is biological insight. This is more difficult to measure or even define, but it represents one of the ultimate goals of genetics: characterizing biological mechanisms to improve prediction and treatment for all. This goal may never be achieved by increasing sample size with shallow phenotyping, because shallow phenotypes are confounded by genetic effects that do not pertain to MDD biology[11]. In contrast, deep phenotyping in clinical cohorts (for example, the Psychiatric Genomics Consortium (PGC)[4] and the China, Oxford and VCU Experimental Research on Genetic Epidemiology (CONVERGE)[9]) has identified a handful of replicated genetic loci that could potentially generate hypotheses on MDD-specific biology. However, this has not yet been demonstrated robustly, as current sample sizes simply do not provide the power to yield enough genetic signals for definitive biological inferences[9].

In this paper, we propose to bridge the shallow–deep gap by integrating information across hundreds of MDD-relevant phenotypes in UK Biobank[11,16] (Fig. 1). We focus on using phenotype imputation[17,18] to increase the effective sample size for the deepest MDD phenotype in UK Biobank (LifetimeMDD)[11], which dramatically improves GWAS power and PRS accuracy over any individual MDD phenotype[19]. We extensively characterize the genetic architecture underlying these imputed phenotypes and show that they remain specific to LifetimeMDD. Further, we develop a novel approach to partly remove nonspecific signals from GWASs on shallow phenotypes akin to latent factor corrections in expression quantitative trait locus (eQTL) studies[20–23]. We also investigate phenotype integration via GWAS summary statistics using multi-trait analysis of GWAS (MTAG), which offers varying specificity and sensitivity depending on input choices. Finally, we developed a novel metric to quantify the specificity of a given PRS, which demonstrates that imputed deep phenotypes of MDD are both more specific and more sensitive than observed shallow phenotypes.

## Results
### Phenotype imputation increases effective sample size
We focused on the deepest available measure of MDD in UK Biobank[11], LifetimeMDD, which we derived by applying clinical diagnostic criteria in silico to MDD symptom data from the Patient Health Questionnaire 9 (PHQ9) and the Composite International Diagnostic Interview Short Form (CIDI-SF) in the online Mental Health Questionnaire (MHQ). This procedure identified 16,297 LifetimeMDD cases and 50,867 controls. Because most individuals did not complete these questionnaires, LifetimeMDD was missing for 269,962 individuals. We also studied a shallow measure of MDD, GPpsy[11], defined as seeking help from a general practitioner (GP) for "depression, anxiety, tension, or nerves". For imputation and downstream analyses, we used a broad depression-relevant phenome with 217 phenotypes, including comorbidities and family

history, as well as socioeconomic, demographic and environmental phenotypes (Supplementary Note and Supplementary Table 1).

We first imputed the depression phenome using SoftImpute[24] (Methods), a variant of principal component analysis (PCA) that identifies latent factors from observed data and uses them to impute missing data. We previously found SoftImpute to be the most scalable among several established approaches[17,24]. We tuned SoftImpute's regularization parameter using realistically held-out test data by taking unions of missingness patterns across samples[17] and also used this approach to estimate the imputation accuracy for each phenotype (Extended Data Fig. 1)[17]. Imputation accuracy varied widely across phenotypes, ranging from $R^2 = 1\%$ for being a twin to $R^2 = 97\%$ for neuroticism score. For LifetimeMDD (80% missing), we estimated the phenotype imputation $R^2$ to be 40%. This roughly translates to doubling its effective sample size[17,25] ($n_{observed} = 67,000$, $n_{effective} = 166,000$; Methods and Extended Data Fig. 1). Imputation accuracy was comparable when stratifying by sex, which is a significant MDD risk factor[26–28] (Supplementary Fig. 1 and Supplementary Table 2). We found that the imputed measures had deflated variances and inflated correlations (Supplementary Note and Supplementary Fig. 2), as expected[17]. This effect could bias some downstream tests, such as tests for genetic correlation. One main goal of this work was to determine whether this approach to phenotype imputation is biased for large-scale single-trait genetic studies.

Finally, we applied a new deep-learning imputation method, Auto-Complete[29], to the same phenotype matrix (Methods). AutoComplete improved estimated imputation accuracy for most phenotypes with >10% missingness (29 of 42) and increased the average estimated by $R^2$ by 2.9%.

### Phenotype imputation improves GWAS power
We performed GWASs on observed LifetimeMDD ($n = 67,164$), imputed values of LifetimeMDD (ImpOnly, $n = 269,962$) and the concatenation of imputed and observed LifetimeMDD (ImpAll, $n = 337,126$; Fig. 2 and Methods). GWAS on the observed values of LifetimeMDD identified one significant locus (Fig. 2e). In GWASs on the imputed values, the number of GWAS loci increased to 13 and 18 for SoftImpute and AutoComplete, respectively (Fig. 2a, b and Supplementary Table 3). Finally, in GWASs on the combination of both imputed and observed values, the number of significant loci further increased to 26 and 40 for SoftImpute and AutoComplete, respectively (Fig. 2c, d and Supplementary Table 3). We confirmed that these improvements in the number of GWAS hits over the single hit from observed LifetimeMDD were unlikely to occur by chance (Supplementary Fig. 3).

We investigated whether the new GWAS hits from phenotype imputation were MDD specific by comparing the ImpOnly GWASs to other MDD GWASs. First, we compared the two imputation methods. Of the 13 and 18 ImpOnly GWAS loci for SoftImpute and AutoComplete, respectively, 8 overlapped (giving a total of 23; Extended Data Fig. 2). Further, 9 of the remaining 15 loci had $P < 10^{-5}$ in both ImpOnly GWASs, and all 15 of the 23 loci had $P < 0.05/23$. Overall, our two imputation methods captured highly overlapping genetic signals, but AutoComplete identified more loci. Next, we assessed the eight shared hits in four nonoverlapping depression cohorts (Methods and Supplementary Note): observed LifetimeMDD in UK Biobank, self-reported depression diagnosis or treatment in 23andMe[7], the 29 MDD cohorts of the PGC[4] (PGC29) and Danish registry data on MDD cases and population controls (iPSYCH[15,30]). For reference, we also compared our findings to those for neuroticism in UK Biobank, a personality trait that is genetically correlated with, but distinct from, MDD[31]. We found that all eight hits shared by both ImpOnly GWASs had sign-consistent effect sizes across all four depression cohorts, as well as in neuroticism. Moreover, all eight hits had $P < 0.05/23$ for observed LifetimeMDD in UK Biobank. Finally, of the 23 single-nucleotide polymorphisms (SNPs) significant in either of the ImpOnly GWASs, 18 replicated in at

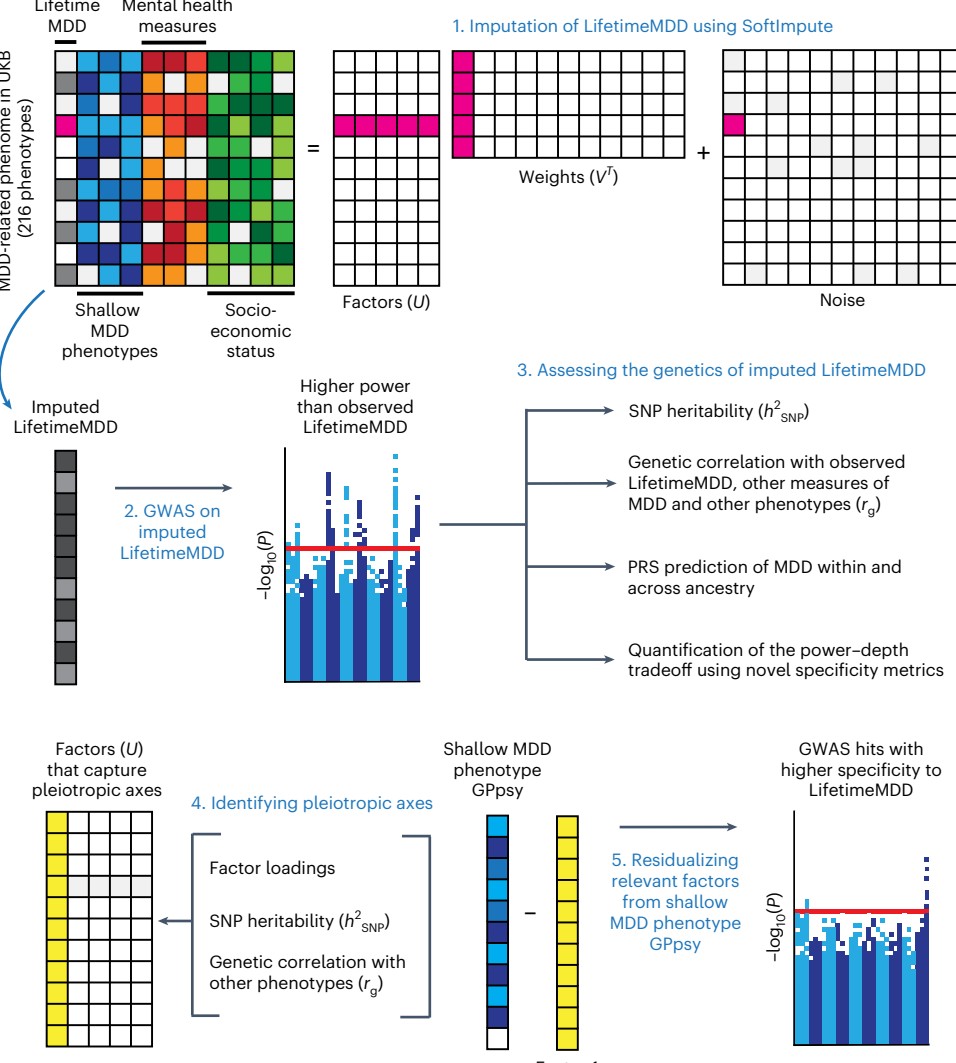

**Fig. 1 | Study overview. (1),** We imputed LifetimeMDD using a partially observed matrix of depression-relevant phenotypes in UK Biobank (UKB). We focused on using SoftImpute, which also produces latent phenome-wide factors. **(2),(3),** We then performed GWASs on observed and imputed values of LifetimeMDD **(2),** as well as downstream polygenic analyses, including in-sample and out-of-sample PRS predictions of MDD **(3). (4),(5),** We also studied the genetic basis of the latent factors of the depression phenome **(4)** and residualized latent factors from shallow MDD phenotypes to remove nonspecific pleiotropic effects **(5).**

least one GWAS of observed MDD at $P < 0.05/23$ (Fig. 2g). Altogether, these results show that the loci underlying imputed LifetimeMDD are relevant to MDD.

We then checked whether the ImpOnly GWAS preserved the polygenic architecture of LifetimeMDD in terms of heritability and genetic correlation. First, we found that the liability-scale SNP-based heritability ($h^2_{g(liab)}$ from LDSC[32]) was lower for imputed LifetimeMDD (Soft-ImpOnly $h^2_{g(liab)} = 13.1\%$, standard error (SE) = 1.0%; Auto-ImpOnly $h^2_{g(liab)} = 14.0\%$, SE = 1.1%) than for observed LifetimeMDD ($h^2_{g(liab)} = 19.0\%$, SE = 2.9%; Fig. 2f). Nonetheless, the genetic correlations between imputed and observed LifetimeMDD were near 1 (Soft-ImpOnly: $r_g = 0.97$, SE = 0.02; Auto-ImpOnly: $r_g = 0.96$, SE = 0.03), as it was between the two imputation methods ($r_g = 1.00$, SE = 0.004). Moreover, the $r_g$ between the ImpOnly phenotypes and other depression phenotypes largely mirrored the $r_g$ based on observed LifetimeMDD (Fig. 2h).

Finally, we tested for effect size heterogeneity between the ImpOnly and observed LifetimeMDD GWASs. We used a random-effect meta-analysis[33] (Methods), as the ImpOnly and observed LifetimeMDD GWASs used nonoverlapping individuals. We found no significant heterogeneity in effect size between ImpOnly and observed LifetimeMDD

at genome-wide significance (Extended Data Fig. 2), and across the 13 and 18 GWAS hits in Soft-ImpOnly and Auto-ImpOnly, respectively, 6 and 4 SNPs showed significant heterogeneity at $P < 0.05/23$. Altogether, our results show that imputed LifetimeMDD is noisier than observed LifetimeMDD but captures similar genetic effects.

## Phenome-wide factors index pleiotropic axes of depression risk

We examined the top latent factors in SoftImpute to understand what phenotypic correlations drive the imputation. We used two statistical metrics to prioritize significant factors. First, we quantified the phenome-wide variance explained (Methods and Fig. 3a): the top handful of factors clearly stood out, with factors becoming comparable to background noise levels around factor 30. Second, we quantified factor stability by calculating the $R^2$ between factors estimated on separate halves of the data, similar to prediction strength in clustering[34] (Methods and Fig. 3b). We found that the first ten factors were very stable (min $R^2$ of ~ 98%), with stability decaying steadily thereafter (factors 11–20, average $R^2$ of ~ 80%; factors 21–30, average $R^2$ of ~60%). We conservatively conclude that the first ten or so factors are statistically meaningful.

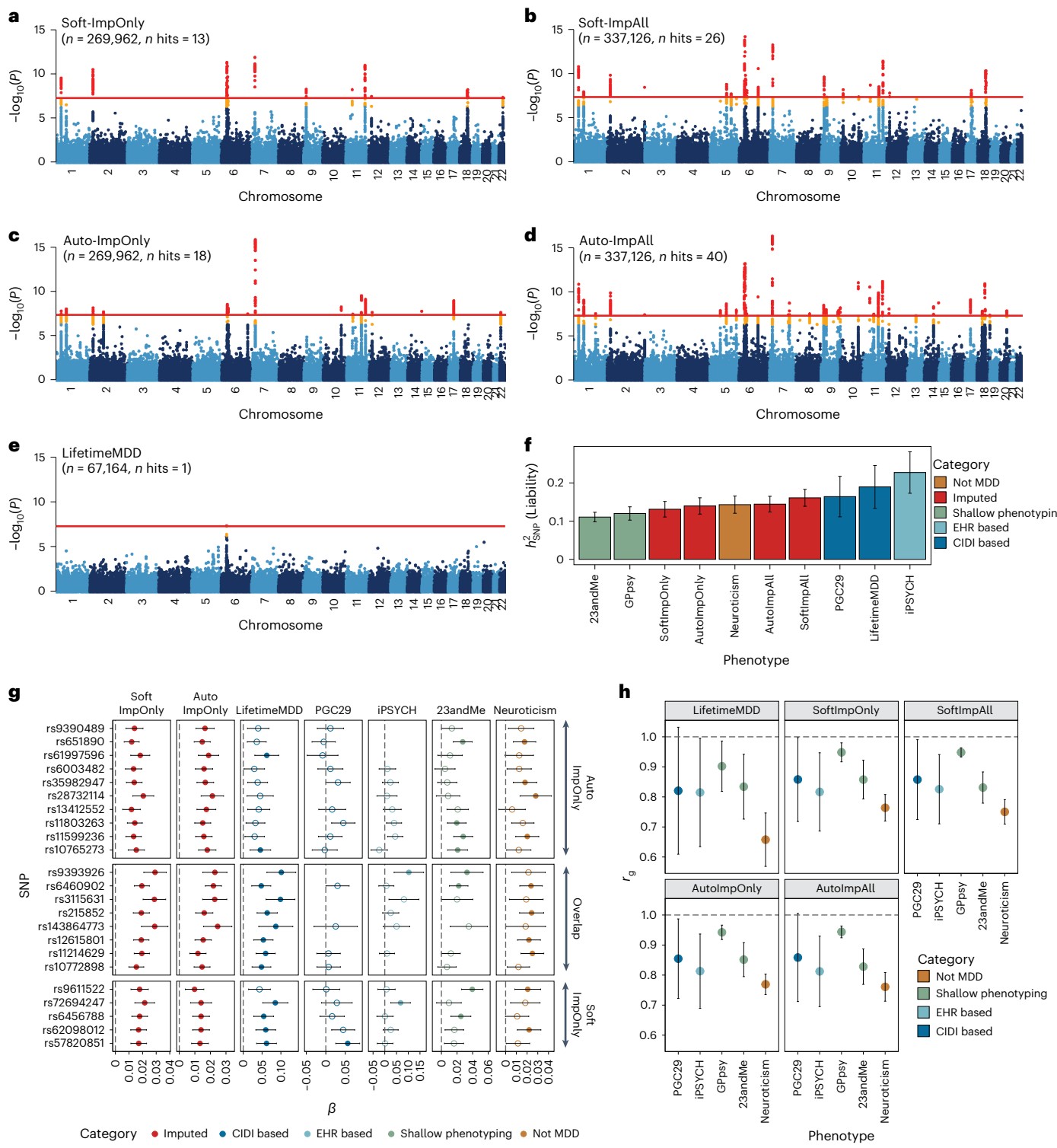

**Fig. 2 | Genetic architecture of observed and imputed LifetimeMDD.**
**a**–**e**, Manhattan plots for linear regression GWASs on imputed LifetimeMDD
values from SoftImpute (**a**) and AutoComplete (**c**) (Soft-ImpOnly and Auto-
ImpOnly, $n = 269,962$) and combined imputed and observed LifetimeMDD values
from SoftImpute (**b**) and AutoComplete (**d**) (Soft-ImpAll and Auto-ImpAll,
$n = 337,126$) and logistic regression GWAS on observed LifetimeMDD (**e**)
($n = 67,164$). $-\log_{10}(P)$ values shown on the $y$ axis were before adjustment for
multiple testing; red lines show the genome-wide significance threshold of

$P < 5 \times 10^{-8}$; $P$ values and test statistics for all GWAS significant SNPs in **a**–**e** are
shown in Supplementary Table 3. **f**, Liability-scale estimates of SNP-based
heritability and **h**, genetic correlation between all UK Biobank phenotypes
($n$ values for Soft, Auto and LifetimeMDD GWASs as above; GPpsy $n = 332,629$;
neuroticism $n = 274,056$) and external MDD studies from PGC ($n = 42,455$),
iPSYCH ($n = 38,128$) and 23andMe ($n = 307,354$). **g**, Replication of GWAS effect
sizes from Soft-ImpOnly and Auto-ImpOnly in observed LifetimeMDD and
external MDD studies. All error bars indicate 95% CI.

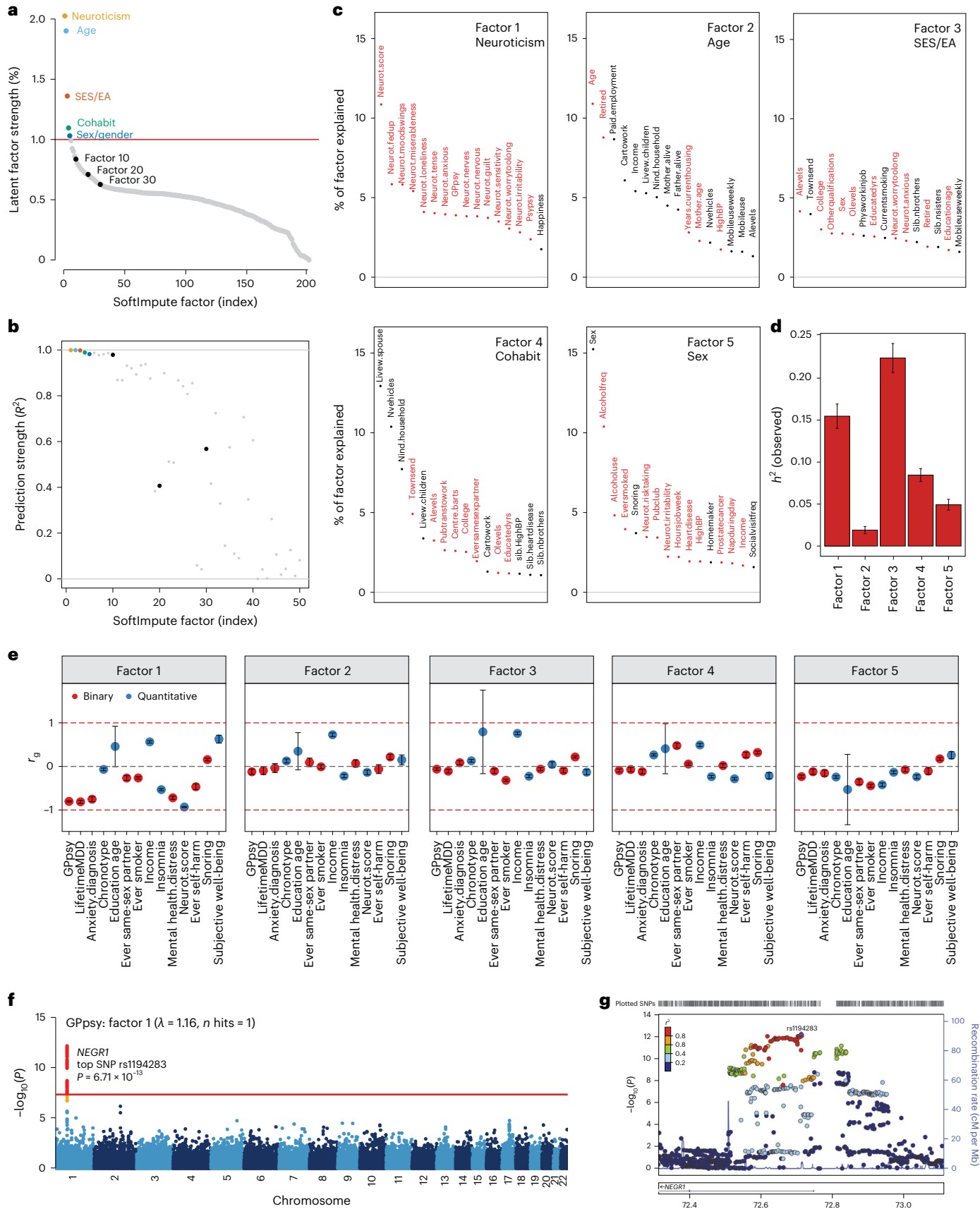

**Fig. 3 | Characterizing top latent factors driving SoftImpute. a,b,** Statistical importance of each factor measured by percentage variance explained in the phenotype matrix (**a**) and factor prediction strength (**b**). **c,** Top phenotype loadings for the top five SoftImpute factors. **d,e,** Estimates of heritability (**d**) (n = 337,126 for all factors) and genetic correlation (**e**) of the top five SoftImpute factors with MDD-relevant traits (n values for all MDD-relevant traits can be found in Supplementary Table 1). **f,** Logistic regression GWAS Manhattan plot of GPpsy conditioning on SoftImpute factor 1; the red line shows the genome-wide significance threshold. **g,** LocusZoom plot of the significant GWAS locus in gene *NEGR1*. All error bars indicate 95% CI.

We studied the genetic basis of each factor with GWASs. The number of GWAS hits ranged from 3 (factor 2, mainly indexing age-related phenotypes; Fig. 3c) to 309 (factor 3, mainly indexing socioeconomic status and education attainment), with $\lambda_{GC}$ ranging from 1.15 (factor 2) to 2.11 (factor 3) (Supplementary Fig. 5). We next estimated heritabilities and found they ranged from $h_g^2 = 1.9\%$ (SE = 0.2%) for factor 2 to $h_g^2 = 22.4\%$ (SE = 0.9%) for factor 3 (Fig. 3d and Supplementary Fig. 4). Finally, we profiled the genetic correlation between factors and MDD-related phenotypes (Fig. 3e and Supplementary Fig. 4). We found that the genetic correlations closely mirrored the factor loadings, which are based on phenotypic correlations. For example, factor 1 had $r_g = -0.93$ (SE = 0.01) with neuroticism and factor 3 was correlated with years of education ($r_g = 0.79$, SE = 0.96) and income ($r_g = 0.75$, SE = 0.03).

Given these results, we hypothesized that our top phenome-wide factors partly capture nonspecific pathways contributing to shallow MDD phenotypes. To test this hypothesis, we performed GWAS analysis on a shallow MDD measure, GPpsy ($n = 332,629$), conditional on factor 1. This is akin to removing confounders such as batch effects in eQTL studies through conditioning on latent factors. We found that only 1 of the 25 GWAS hits for GPpsy remained after adjusting for factor 1 (Fig. 3f). This hit overlaps *NEGR1* (top SNP rs1194283, odds ratio (OR) = 1.05, SE = 0.0065, $P = 6.71 \times 10^{-13}$; Fig. 3g), which has been identified as an MDD risk locus in multiple GWASs with varying phenotyping approaches[4,6,8,35–37]. Intriguingly, this locus also has replicated associations with body mass index and obesity in multiple populations[38–41]. We also performed GWASs conditioned on each of the other top ten factors, which generally had little impact (Supplementary Fig. 6). One clear exception, however, was that adjusting for factor 3 increased the number of GPpsy GWAS hits from 25 to 35. These additional loci could be false positives from collider bias[23,42,43].

## MTAG improves GWAS power but is sensitive to inputs

As an alternative to phenotype imputation, we next evaluated phenotype integration at the GWAS summary statistic level using MTAG[19], an inverse-covariance-weighted meta-analysis for GWAS on multiple traits. We did not use all 217 phenotypes in MTAG for two reasons. First, MTAG requires GWAS to be run on each input phenotype, which is computationally intractable for hundreds of phenotypes. Second, MTAG produces false positives when applied to large numbers of input GWAS[19]. Instead, we evaluated six different sets of input GWAS to MTAG, producing six different integrated LifetimeMDD GWASs (Fig. 4a, Extended Data Fig. 3, Supplementary Table 4 and Supplementary Note).

All six MTAG input choices increased the number of GWAS hits from observed LifetimeMDD. On the low end, MTAG using family history measures of depression yielded five GWAS hits (MTAG.FamilyHistory, $\lambda_{GC} = 1.20$; Fig. 4b). On the high end, MTAG using shallow MDD phenotypes and environmental factors (such as recent stressful life events, lifetime traumatic experiences and the Townsend deprivation index) yielded 33 GWAS hits (MTAG.All, $\lambda_{GC} = 1.45$; Fig. 4c). We note that MTAG.AllDep is analogous to prior depression phenotypes defined by manually combining similar depression measures[14] and that MTAG.FamilyHistory is analogous to prior methods that integrated family history into GWAS analysis[44,45]. Of the 51 total hits across all MTAG runs, 34 overlapped hits from the imputed GWASs (Extended Data Fig. 3). Notably, we found that adding more input phenotypes in MTAG always increased the number of GWAS hits. This reflects a combination of added power by leveraging pleiotropy and an increased false positive rate[19]. Additionally, MTAG GWASs yielded substantially inflated heritability estimates (on both the liability and observed scales), which increased with more input GWAS. For example, MTAG.All gave $h_{g(liab)}^2 = 84.9\%$ (SE = 5.6%), compared to $h_{g(liab)}^2 = 19.0\%$ (SE = 2.9%) for observed LifetimeMDD (Fig. 4d and Supplementary Table 5).

We next examined genetic correlations between MTAG and other MDD GWASs (Fig. 4e). First, MTAG.All, which included the most input GWAS, clustered together with the imputed GWASs, which leveraged

all 217 phenotypes. Second, MTAG using shallow MDD phenotypes (MTAG.AllDep and MTAG.GPpsy) clustered with the GWAS on GPpsy. Third, neuroticism was significantly more genetically correlated with MTAG.Envs ($r_g = 0.84$, SE = 0.01) than LifetimeMDD ($r_g = 0.66$, SE = 0.06). These results are consistent with prior observations that MTAG-based summary statistics modestly inflate genetic correlation with the input phenotypes[46]. Overall, the genetic correlations between MTAG and LifetimeMDD were high, with the lowest value derived for MTAG.Envs ($r_g = 0.90$, SE = 0.03).

Finally, to compare like to like, we evaluated SoftImpute's accuracy using only the MTAG.All input phenotypes (plus sex, age and 20 PCs). We found that imputation performed much worse with this reduced set of phenotypes (phenome-wide $R^2$ decreased from 59.6% to 39.5%, $P < 2 \times 10^{-5}$, pooled $t$-test across folds; Supplementary Fig. 2 and Supplementary Table 2).

## Phenotype imputation and MTAG improves PRS accuracy

We then assessed the within-sample prediction accuracy of PRSs built from integrated MDD GWAS. We used ten-fold cross-validation to obtain the Nagelkerke's $R^2$ prediction accuracy for LifetimeMDD in white British individuals in UK Biobank. We jointly cross-validated the phenotype imputation and PRS construction (Methods). For MTAG, we jointly cross-validated the GWASs on secondary input phenotypes. For context, we compared these PRSs to ones built from observed LifetimeMDD ($n = 67,164$; for $n_{effect}$ see Supplementary Table 6) and GPpsy ($n = 332,629$) in UK Biobank[11], MDD defined by structured interviews in PGC29 (ref. 4) ($n = 42,455$), affective disorder defined by Danish health registries in iPSYCH[15] ($n = 38,123$) and self-reported depression in 23andMe[7] ($n = 307,354$; Supplementary Table 6).

We found that imputing LifetimeMDD doubled PRS prediction accuracy over observed LifetimeMDD (Fig. 5a; LifetimeMDD: $R^2 = 1.0\%$, 95% confidence interval (CI) = 0.6–1.4%; Soft-ImpAll: $R^2 = 2.1\%$, 95% CI = 1.3–2.9%; Auto-ImpAll: $R^2 = 2.2\%$, 95% CI = 1.4–3.0%). Consistent with prior reports[11,12], we found that the GPpsy PRS predicted LifetimeMDD better than the LifetimeMDD PRS itself ($R^2 = 1.6\%$, 95% CI = 0.6%–2.4%) because it has a roughly four times larger sample size. Nonetheless, both the SoftImpute and AutoComplete PRSs outperformed the GPpsy PRS, demonstrating that integrating shallow and deep phenotypes through imputation can improve PRS accuracy over that with either class of phenotypes alone. Finally, we found that the imputed LifetimeMDD PRS substantially outperformed the PRSs from iPSYCH ($R^2 = 0.6\%$, 95% CI = 0.2–0.9%) and 23andMe ($R^2 = 1.3\%$, 95% CI = 0.7–1.9%), even though iPSYCH used deeper phenotypes and 23andMe had a large sample size.

The performance of the MTAG PRSs mirrored the MTAG GWAS results and depended on the input phenotypes (Fig. 4 and Extended Data Fig. 3). First, MTAG.FamilyHistory showed improved accuracy compared to observed LifetimeMDD but underperformed in comparison to the imputed PRSs ($R^2 = 1.5\%$, 95% CI = 0.6–2.5%; Fig. 5a). On the other hand, MTAG.All outperformed the imputed PRSs by about 20% ($R^2 = 2.6\%$, 95% CI = [1.3, 3.9%]; Fig. 5a). In particular, this demonstrates that MTAG with more than ten inputs, which is nonstandard and likely yields miscalibrated GWAS results, can significantly improve PRS prediction.

## Phenotype imputation and MTAG improves PRS portability

We next asked whether phenotype integration improves PRS predictions in different cohorts, diagnostic systems and/or populations. Demonstrating portability is essential to establish that phenotype integration does not merely reflect dataset-specific biases.

First, we tested PRS accuracy in non-British European-ancestry individuals in UK Biobank (UKB.EUR, $n = 10,166$). These individuals were measured on the same LifetimeMDD phenotype as our sample of white British UK Biobank individuals and also have European ancestry, and hence represent the most similar cohort (Supplementary

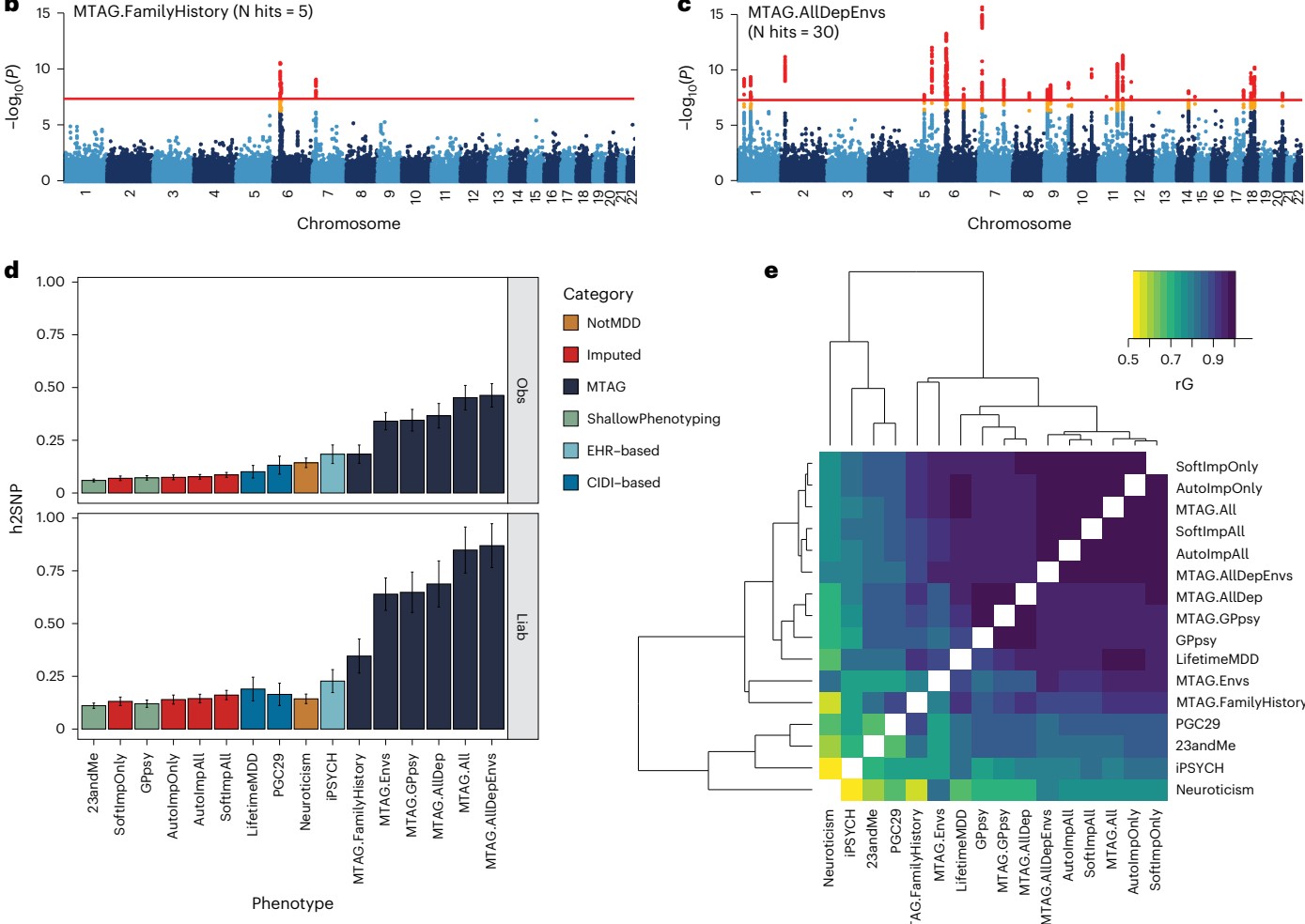

**a**

| MTAG run | Phenotypes | MTAG mean $\chi^2$ | MTAG neff | MTAG nhits |
|---|---|---|---|---|
| MTAG.FamilyHistory | Mother.severedepression, father.severedepression, sibling .severedepression | 1.196 | 111,263 | 5 |
| MTAG.Envs | Townsend, neuroticismscore.baseline, trauma, stressbinary | 1.343 | 194,650 | 17 |
| MTAG.GPpsy | GPpsy | 1.356 | 201,743 | 19 |
| MTAG.AllDep | GPpsy, Psypsy, DepAll, SelfRepDep, ICD10Dep | 1.386 | 224,871 | 22 |
| MTAG.All | GPpsy, Psypsy, DepAll, SelfRepDep, ICD10Dep, townsend, neuroticismscore.baseline, trauma, stressbinary, mother.severedepression, father.severedepression, sibling.severedepression | 1.45 | 262,199 | 33 |
| MTAG.AllDep+Envs | GPpsy, Psypsy, DepAll, SelfRepDep, ICD10Dep, townsend, neuroticismscore.baseline, trauma, stressbinary | 1.485 | 282,558 | 30 |

**Fig. 4 | MTAG results for different choices of input phenotypes. a**, Description of the evaluated input choices for MTAG and their resulting GWAS summary statistics. The MTAG effective sample size refers to the power-equivalent sample size of the MTAG GWAS versus the single-trait GWAS[19], calculated as $n_{effect} = n_{single} \times (\chi^2_{MTAG} - 1)/(\chi^2_{single} - 1)$, where the $\chi^2$ terms are the average GWAS chi-squared values. **b,c**, Manhattan plots for the MTAG models with the fewest (MTAG.FamilyHistory) (**b**) and most (MTAG.AllDep+Envs) (**c**) GWAS hits. $-\log_{10}(P)$ values shown on the *y* axis were before adjustment for multiple testing; red lines show the genome-wide significance threshold of $P < 5 \times 10^{-8}$. **d**, SNP-based heritability estimates on the observed and liability scales for observed, imputed, and MTAG GWASs on LifetimeMDD as well as reference phenotypes ($n_{effect}$ is shown in Supplementary Table 5). **e**, Estimated genetic correlations for observed, imputed and MTAG analyses of LifetimeMDD and reference phenotypes. All error bars indicate 95% CI.

Note and Supplementary Fig. 7). Although the small sample size limits definitive conclusions, we observed a nearly identical pattern among the PRS methods as in our training sample: imputation and MTAG almost always improved prediction accuracy compared to both LifetimeMDD and GPpsy (Fig. 5b). We next assessed portability to two large

European-ancestry cohorts from iPSYCH (2012 cohort ($n = 42,250$) and 2015i cohort ($n = 23,351$); Supplementary Note). These nonoverlapping samples were drawn from a nationwide Danish birth cohort with diagnoses obtained from national health registers[30,47]. We again found qualitatively identical results, with imputation outperforming both

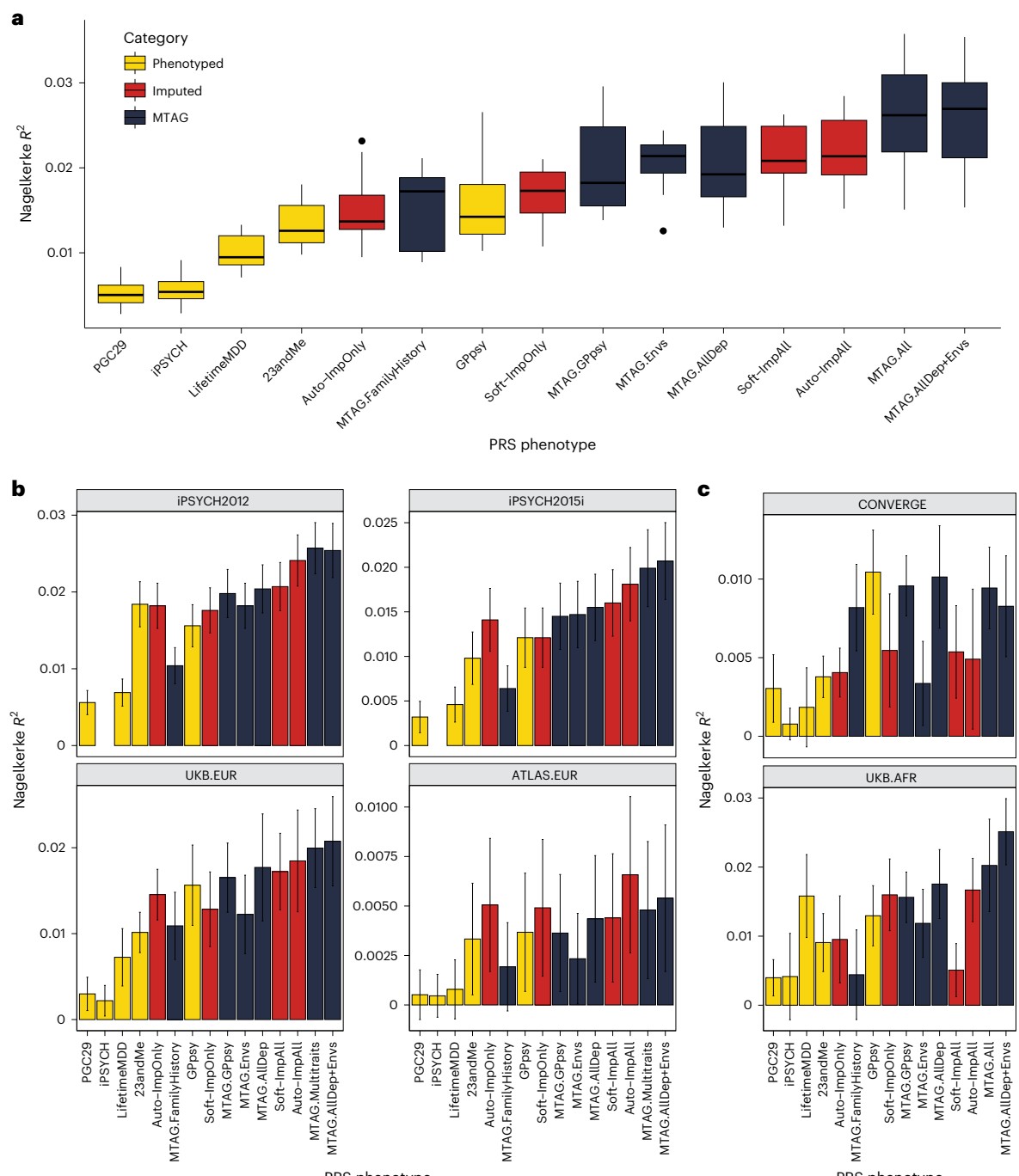

**Fig. 5 | PRS performance using observed, imputed and/or meta-analyzed MDD. a**, PRS prediction accuracy for LifetimeMDD ($n$ = 67,164) in unrelated white British individuals in UK Biobank using ten-fold cross-validation. For the imputed PRSs, we also cross-validated the imputation. Median values are shown as a line in the box; whiskers of boxplots are 1.5 times the interquartile range; outliers outside of the interquartile range are shown as filled dots. **b**, Out-of-sample PRS prediction accuracy in four additional cohorts with European ancestries (iPSYCH2012, $n$ = 42,250; iPSYCH2015i, $n$ = 23,351; UKB.EUR, $n$ = 10,193; ATLAS.EUR, $n$ = 14,366). **c**, PRS prediction accuracy in African-ancestry individuals in UK Biobank ($n$ = 687) and Han Chinese-ancestry individuals in CONVERGE ($n$ = 10,502). All error bars indicate 95% CI.

LifetimeMDD and GPpsy and the best MTAG setting outperforming imputation (Fig. 5b). Finally, we tested portability to European-ancestry individuals in the ATLAS dataset based on MDD as defined in University of California - Los Angeles electronic health record (EHR) data[48,49] (ATLAS.EUR, $n$ = 14,388; Supplementary Note, Supplementary Fig. 8 and Supplementary Tables 6–8). Again, the small sample size prevented definitive comparisons, but phenotype imputation and the best MTAG setting improved estimated accuracy (Fig. 5b).

We next tested these PRSs in individuals with non-European genetic ancestries, including African-ancestry individuals[7] in UK Biobank with observed LifetimeMDD status (UKB.AFR, $n$ = 687), as well as Han Chinese-ancestry individuals in the CONVERGE cohort[9,50] ($n$ = 10,502; Supplementary Note) who were assessed for severe, recurrent MDD (Fig. 5c, Supplementary Note and Supplementary Table 6). Consistent with previous studies[51–53], we found that the PRSs we derived from GWASs on European-ancestry cohorts generally had

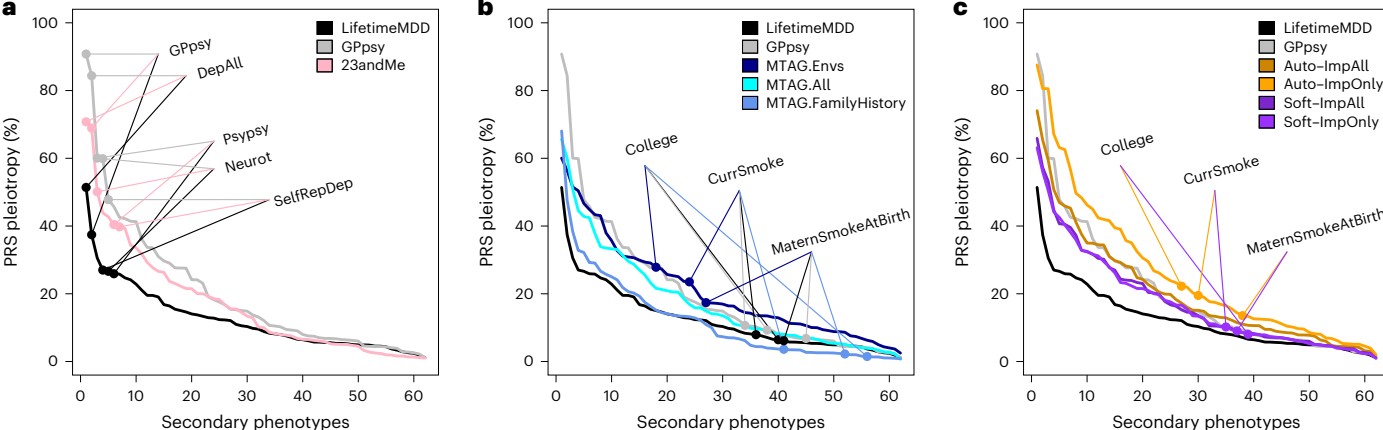

**Fig. 6 | Phenome-wide PRS pleiotropy quantifies nonspecificity.** PRS pleiotropy spectra across the depression-relevant phenome, defined as the ratio of PRS prediction accuracy for secondary traits relative to LifetimeMDD (PRS pleiotropy = $R^2_{secondary}/R^2_{LifetimeMDD}$). **a**, PRSs derived from GWAS on shallow MDD phenotypes (GPpsy or 23andMe) are less specific to LifetimeMDD than the PRS derived from the GWASs on LifetimeMDD. **b**, MTAG-based PRSs range from highly specific (MTAG.FamilyHistory) to less specific (MTAG.Envs) compared to PRSs derived from shallow MDD phenotypes. **c**, SoftImpute PRSs are more specific than PRSs derived from shallow MDD phenotypes, while AutoComplete PRSs are similar. Note that GPpsy and LifetimeMDD are each used in two ways: (1) to build the PRS and (2) to evaluate PRS pleiotropy.

poorer portability to non-European-ancestry cohorts, making firm conclusions difficult. Nonetheless, the best MTAG setting was always nearly optimal, and PRSs based on imputed LifetimeMDD always outperformed the PRS based on observed LifetimeMDD. Finally, we tested PRS prediction accuracy in UK Biobank individuals with Asian ancestry (UKB.ASN, $n = 334$), as well as ATLAS individuals who self-identified as Latino (ATLAS.LAT, $n = 2,454$), Black (ATLAS.AFR, $n = 1,158$) or Asian (ATLAS.ASN, $n = 1,996$). However, the power was too low in these small cohorts for meaningful interpretation (Supplementary Fig. 9).

### A new metric for the specificity of PRSs

While phenotype integration improves PRS prediction in UK Biobank and in external cohorts, this may come at the cost of reduced specificity to MDD. This is because integration explicitly borrows information from secondary phenotypes, which could introduce genetic signals that are not specific to MDD. To quantify this spillover of nonspecific effects into an MDD PRS, we compared prediction accuracy for LifetimeMDD to prediction accuracy for secondary phenotypes. We call this metric of specificity 'PRS pleiotropy' ($R^2_{secondary}/R^2_{LifetimeMDD}$). Because core MDD biology is likely partly pleiotropic, its PRS pleiotropy should be nonzero for many secondary phenotypes. We further expect that shallow MDD phenotypes, such as GPpsy, would generally have higher PRS pleiotropy than LifetimeMDD[11].

For all PRSs based on observed and integrated MDD, we calculated PRS pleiotropy for 172 secondary phenotypes used in imputation. We then investigated the 62 secondary phenotypes that were significantly predicted by any examined PRS ($P < 0.05/172$; Methods). Visualizing PRS pleiotropy for observed LifetimeMDD across this depression phenome showed a spectrum of highly linked traits, including shallow MDD phenotypes such as GPpsy and genetically correlated traits such as neuroticism, that quickly faded across successive phenotypes (Fig. 6a). By comparison, GPpsy broadly had higher PRS pleiotropy across secondary phenotypes, indicating that GPpsy captures less-specific biology than LifetimeMDD, as expected. We also found that the 23andMe GWAS had similar PRS pleiotropy to GPpsy, consistent with the fact that both measure MDD by self-reported depression.

Our main question here was whether phenotype integration has more PRS pleiotropy than self-reported depression as in GPpsy and 23andMe. We first evaluated PRS pleiotropy for MTAG and found that specificity highly depended on the input GWAS (Fig. 6b). First, MTAG.Envs had far higher PRS pleiotropy than GPpsy, showing that its high

PRS power comes at a high cost to specificity. On the other hand, MTAG.All had a similar PRS pleiotropy as GPpsy and nearly double the PRS $R^2$; hence, MTAG.All is clearly superior to GPpsy. MTAG.FamilyHistory had the opposite properties: it only modestly improved PRS $R^2$ over observed LifetimeMDD, but this benefit came without loss of specificity. We then evaluated PRS pleiotropy for imputed phenotypes (Fig. 6c). The SoftImpute ImpAll and ImpOnly PRSs were both more specific to LifetimeMDD than the GPpsy PRS, which is notable given that the imputed values were constructed from more than 200 phenotypes, including GPpsy. The AutoComplete ImpOnly PRS was less specific than GPpsy, although the ImpAll PRS was comparable.

Finally, we evaluated PRS pleiotropy relative to observed LifetimeMDD to ask which of the phenotypes had excess pleiotropy in the integrated PRSs. We defined 'excess PRS pleiotropy' as the PRS pleiotropy minus the PRS pleiotropy of observed LifetimeMDD[11]. As expected, self-reported depression PRSs (GPpsy and 23andMe) had high excess PRS pleiotropy, especially when compared to shallow MDD measures (Extended Data Fig. 4a). Likewise, MTAG.Envs had substantial excess PRS pleiotropy, especially for socioeconomic measures such as years of education (Extended Data Fig. 4b). Notably, MTAG.FamilyHistory had far less excess PRS pleiotropy than other MTAG settings or GPpsy, as well as reduced pleiotropy for several socioeconomic measures. Finally, SoftImp-All had lower excess PRS pleiotropy than GPpsy (41/62 phenotypes); however, AutoImp-All had higher excess PRS pleiotropy (Extended Data Fig. 4c). Overall, MTAG can outperform imputation in PRS sensitivity or specificity depending on input, while imputation provides a simple and scalable approach that performs well in both power and specificity.

We then downsampled each GWAS used to build a PRS to assess the impact of sample size (Supplementary Note). Overall, we found that PRS pleiotropy was stable, although it can be upwardly biased for sample sizes below 100,000 (due to low power). In particular, our results were robust to differences in the training PRS sample sizes, except that observed LifetimeMDD PRS pleiotropy is a conservative baseline because it is trained on 67,000 individuals (Extended Data Fig. 5 and Supplementary Figs. 10 and 11). Further, we confirmed that these results persisted when we used the same SNPs in each PRS (Supplementary Note and Extended Data Fig. 6).

### Discussion

In this paper, we address the power–specificity tradeoff between deep and shallow MDD phenotypes by integrating them using phenotype

imputation or MTAG. We show that the integrated MDD phenotypes greatly improve GWAS power and PRS accuracy while, crucially, preserving the genetic architecture of MDD. We propose a novel metric to assess the disorder specificity of a PRS that is widely applicable to biobank-based GWASs. This metric characterizes a power–specificity tradeoff. For MTAG, adding more phenotypes increases power but generally sacrifices specificity. Imputing LifetimeMDD with SoftImpute, on the other hand, better preserves specificity. Overall, both approaches to phenotype integration outperform either deep or shallow phenotypes alone.

Phenotype imputation is a simple and scalable approach that should be considered for most biobank-based genetic studies. An important but challenging future application will be longitudinal data, which are often sporadically measured across time and individuals with potentially extreme nonrandom missingness. One limitation of our specific imputation approaches is that they distort higher-order moments (Supplementary Fig. 1), which will bias some downstream analyses such as genetic correlation. In future work, this could be addressed with multiple imputation[54] or with heteroskedasticity-aware downstream tests[55].

MTAG has several important strengths that complement imputation. First, it operates on summary statistics, which enables incorporating external GWASs and is computationally cheap once GWASs have been performed. Second, we found that MTAG.All generally outperforms imputation in GWAS hits and PRS power and portability. However, MTAG generally has less specificity. A striking exception is MTAG.FamilyHistory, consistent with prior methods that specifically leverage family history to improve genetic studies[44,45,56]. However, MTAG is highly sensitive to input phenotype selection and therefore requires extensive domain knowledge, similar to other work combining multiple depression measures[14]. There are several extensions to further improve MTAG for phenotype integration. First, we could incorporate local estimates of genetic correlation in MTAG, which has improved LifetimeMDD PRSs in UK Biobank[57]. Second, we could directly combine PRSs for multiple traits using weights to optimize prediction[58–60]. Third, we could develop a more systematic approach to choosing MTAG inputs, but this search is limited by the computational cost of performing cross-validated GWASs on each considered trait. Fourth, parametric models of confounding in summary statistics, such as GWAS-by-subtraction[61], could improve specificity. However, these models rely on choosing appropriate inputs and causal models, which is not straightforward for heterogeneous disorders. Finally, we could input the GWASs on imputed phenotypes to MTAG; however, this may exacerbate their biases because imputation inflates the correlations between traits that MTAG leverages (Supplementary Fig. 1).

Our study has implications for improving disorder specificity in future MDD studies. We have worked on the deepest MDD phenotype in UK Biobank, LifetimeMDD, which is derived by applying DSM-5 criteria in silico to self-rated MDD symptoms in the MHQ. Although it lies on essentially the same genetic liability continuum as gold-standard MDD[11], LifetimeMDD is shallow compared to a clinical diagnosis based on a structured in-person interview due to self-report biases and misdiagnoses[62–65]. In the future, this bias could be mitigated using probability weights, which have recently been used for GWASs[66,67]. More broadly, the DSM-5 criteria for MDD have substantial shortcomings in reliability[68–70]. Nonetheless, improving the MDD diagnostic criteria may be only achievable through epistemic iterations[64], a series of efforts to characterize specific genetic signals for the deepest available MDD definition and, in turn, refine our definition of MDD. Our efforts to improve GWAS power and specificity in noisy biobanks advance this process.

Our implementation of phenotype integration uses shallow MDD phenotypes to improve power for LifetimeMDD, and as such its specificity is limited by the specificity of LifetimeMDD. Future statistical methods could go further and improve specificity over existing phenotypes. We have taken a step in this direction with SoftImpute factors,

which revealed a specific locus from a shallow phenotype. This is akin to latent factor correction in genomic studies[23,71–74], and it could be adapted to AutoComplete using, for example, Integrated Gradients methods[75]. However, it is challenging to remove nonspecific signals without removing specific signals or, worse, introducing artificial signals due to collider bias[23,42,76]. This is especially true for disorders such as MDD where epistemic uncertainty clouds what signals are most biomedically useful.

Phenotype integration is broadly applicable to biobank-based genetic studies, which often evaluate a mixture of biomarker-, nurse-, GP-, specialist- and/or patient-defined disorder measures. Further, biobanks offer diverse disorder-relevant phenotypes, such as age of onset, medical procedures, prescriptions, environmental risk factors, family history and socioeconomic measures. While the degree of specificity of phenotype integration will vary between applications, this can be assessed with our novel PRS pleiotropy metric. This empirical measure complements prior theoretical derivations of power–specificity tradeoffs in the meta-analysis of heterogeneous traits[77]. In the future, PRS pleiotropy can be used to evaluate the specificity of newly constructed phenotypes.

Our results have complex implications for equity in genetic studies and clinical care. On the one hand, EHR-derived phenotypes have a history of exacerbating inequities that continues to this day[78–81]. Moreover, phenotype integration uses a reference phenome, which has the potential to propagate systematic biases present in biobank data. Although we found that phenotype integration can improve PRS portability across ancestries, the results are less statistically clear than those in European ancestries. This highlights the need for greater sample sizes in diverse ancestries and better methods for cross-ancestry portability. Careful extensions of our approach, such as residualizing phenome-wide factors, have the potential to improve portability by eliminating confounders. Given the extreme Eurocentric biases in available genomics data, these and other statistical approaches to improve the utility of PRSs for all people are urgently needed.

## Online content

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

## Methods

### Ethical approval

This research was conducted under the ethical approval from the UK Biobank resource under application no. 28709 and 33217. The use of iPSYCH data follows the standards of the Danish Scientific Ethics Committee, the Danish Health Data Authority, the Danish Data Protection Agency and the Danish Neonatal Screening Biobank Steering Committee. Data access was via secure portals in accordance with Danish data protection guidelines set by the Danish Data Protection Agency, the Danish Health Data Authority and Statistics Denmark. Retrospective data collection and analysis for ATLAS was approved by the UCLA Institutional Review Board (IRB)[48]. Patient Recruitment and Sample Collection for Precision Health Activities at UCLA" is an approved study by the UCLA IRB (UCLA IRB17-001013). All necessary patient/participant consent has been obtained and the appropriate institutional forms have been archived. The CONVERGE study was approved by the ethical review boards of Oxford University and participating hospitals. All participants provided written informed consent[9].

### Phenotypes used in phenotype imputation

We considered 217 relevant phenotypes to impute LifetimeMDD[11] in 337,127 individuals of white British ancestry in UK Biobank (Supplementary Table 1). These included (1) LifetimeMDD as defined in Cai et al.[11]; (2) minimal phenotyping definitions of depression based on help-seeking, symptoms, self-reports and/or EHRs as defined in Cai et al.[11]; (3) individual lifetime and current MDD symptoms from the CIDI-SF[82] and PHQ9 from which we derived LifetimeMDD; (4) psychosocial factors; (5) self-reported comorbidities; (6) family history of common diseases; (7) early life factors; (8) socioeconomic phenotypes; (9) lifestyle and environment phenotypes; (10) social support status; and (11) demographic features including age, sex, UK Biobank assessment center as a proxy for geographical residence and 20 genetic PCs. These phenotypes were selected based on their established relevance to MDD and were all collected through either a touchscreen questionnaire completed at the assessment center or the online mental health follow-up questionnaire (MHQ). All UK Biobank data fields, sample sizes and prevalence of binary outcomes are detailed in Supplementary Table 1, and we report levels of missingness for all inputs for multi-phenotype imputation in Extended Data Fig. 1. For PRS pleiotropy analyses, we excluded the 20 genetic PCs, 22 assessment centers and the genotyping array.

### Phenotype imputation with SoftImpute

We fit SoftImpute with the ALS method[24] on the 217 phenotypes making up the MDD-related phenome in UK Biobank, using cross-validation to optimize the nuclear norm regularization parameter. We used our prior approach to make the cross-validation more realistic by copying real missingness patterns instead of completely random entries[17,83], which provides more realistic estimates of imputation accuracy (Extended Data Fig. 1). We previously studied SoftImpute at a smaller scale in comprehensive simulations and several real datasets[17], and we have since used it in several larger studies[17,83,84]. Overall, SoftImpute is extremely simple, robust and scalable. We summarize the SoftImpute model fit by the latent factors (Fig. 3c) and the variances they explain (Fig. 3a), which are akin to the eigenvectors (or PCs) and eigenvalues of the phenotype covariance matrix, respectively. We also estimate the prediction strength (Fig. 3b), which is the squared correlation between two latent factors estimated after splitting the sample into two nonoverlapping halves ($R^2$). We define the effective sample size after-imputation as $N_{obs} + N_{miss} \times R^2$, analogous to genotype imputation[17,18,25]; we note that this approximates the power-equivalent number of observed phenotypes.

### Phenotype imputation with AutoComplete

We developed a new deep learning-based method, AutoComplete, in a companion paper[29]. AutoComplete consists of several fully connected layers with nonlinearities and learns to optimize reconstruction of realistically held-out missing entries. The model is fully differentiable and is fit using stochastic gradient descent. Unlike SoftImpute, AutoComplete's objective function models binary phenotypes. As with SoftImpute, the hyperparameters for AutoComplete were determined through cross-validation on realistically held-out missing data. In this paper, we focus on its application to imputing LifetimeMDD.

### GWASs on observed or imputed phenotypes

GWASs on directly phenotyped and imputed phenotypes in UK Biobank were performed using imputed genotype data at 5,781,354 SNPs (minor allele frequency > 5%, INFO score > 0.9) using logistic regression and linear regression implemented in PLINK v2 (ref. 85) for binary and quantitative traits, respectively. We used 20 PCs computed with flashPCA[86] on 337,129 white British individuals in UK Biobank and genotyping arrays as covariates for all GWASs (see Supplementary Note for details of sample and genotype quality control in UK Biobank). To test for heterogeneity between genetic effects found in GWASs on observed LifetimeMDD and imputed measures of MDD from SoftImpute (Soft-ImpOnly) and AutoComplete (Auto-ImpOnly), we performed a random-effect meta-analysis using METASOFT[33] and tested for heterogeneity between effect sizes at each SNP.

### SNP-based heritability and genetic correlation

To test for SNP-based heritability of each phenotype and the genetic correlation between pairs of phenotypes, linkage disequilibrium (LD) score regression implemented in LDSC v1.0.11 (refs. 32,87) was performed on the GWAS summary statistics using in-sample LD scores estimated in 10,000 random white British UK Biobank individuals at SNPs with minor allele frequency > 5% as reference. For MTAG results, we used the effective sample size estimated in MTAG as the sample size entry in LDSC; for all other GWASs, we used the actual sample size. When we estimated the liability-scale heritability, we assumed the population prevalence of binary phenotypes equaled their prevalence in UK Biobank. For all GWASs, we have indicated their effective sample sizes accounting for imbalance between cases and controls ($n_{effective} = 4/(1/n_{cases} + 1/n_{controls})$) in Supplementary Tables 5 and 6. We note that this is different from the imputation-related definition of effective sample size and also differs from MTAG's definition.

### In-sample PRS prediction of phenotypes in UK Biobank with ten-fold cross-validation

We performed SoftImpute[24] and AutoComplete imputations ten times. Each time we used 90% of the individuals in the input phenotype matrix and then built the PRS from the GWAS results with PRSice v2 (ref. 88) and evaluated predictive accuracy for observed LifetimeMDD and the depression-related phenome (217 phenotypes, used as input in imputation) in the 10% of individuals who were held out. For MTAG[19], we performed a GWAS on each set of input phenotypes (as shown in Fig. 4) ten times, each time using 90% of the individuals in UK Biobank. We then ran MTAG on the GWAS summary statistics on this 90%, built a PRS from the resulting MTAG summary statistics with PRSice v2 and evaluated predictive accuracy for observed LifetimeMDD in the 10% of individuals who were held out. For all PRS predictions, we used 20 genomic PCs and the genotyping array used as covariates. For binary phenotypes, including LifetimeMDD, we evaluated accuracy using Nagelkerke's $R^2$. For all quantitative phenotypes, including neuroticism, we evaluated accuracy using ordinary $R^2$.

### PRS prediction of phenotypes in UK Biobank from external GWAS summary statistics

We constructed PRSs from MDD GWAS summary statistics from PGC29 (ref. 4), iPSYCH[15] and 23andMe[7], as detailed in Supplementary Table 6, and predicted phenotypes in UK Biobank using PRSice v2, with 20 genomic PCs and the genotyping array used in UK Biobank as covariates. For each of these studies, we used only SNPs with imputation INFO

score > 0.9 and minor allele frequency > 5% for constructing the PRS. For binary phenotypes, including LifetimeMDD, we evaluated accuracy using Nagelkerke's $R^2$. For all quantitative phenotypes, including neuroticism, we evaluated accuracy using ordinary $R^2$. We calculated PRS pleiotropy on all secondary phenotypes (not including LifetimeMDD) used in imputation, except for 20 PCs, array and assessment center (total of 172 phenotypes).

## Reporting summary

Further information on research design is available in the Nature Portfolio Reporting Summary linked to this article.

## Data availability

UK Biobank genotype and phenotype data used in this study are from the full release (imputation version 2) of the UK Biobank Resource obtained under application no. 28709 and 33217. We used publicly available summary statistics from PGC29 and 23andMe from the Psychiatric Genomics Consortium (https://www.med.unc.edu/pgc/results-and-downloads), as well as summary statistics for affective disorders in the iPSYCH2012 cohort (https://doi.org/10.6084/m9.figshare.20517330), with references in Supplementary Table 4. The individual-level CONVERGE, Danish and UCLA datasets are not publicly available due to institutional restrictions on data sharing and privacy concerns. Summary statistics of all GWASs used in this paper are available at: https://doi.org/10.6084/m9.figshare.19604335. PRSs for all imputed and MTAG GWASs in this study have been submitted to the PGS Catalogue under publication ID PGP000461 and have been assigned score IDs from PGS003576 to PGS003585.

## Code availability

Publicly available tools that were used in data analyses are described wherever relevant in the Methods and Reporting Summary. Custom code for SoftImpute imputation of the MDD-relevant phenome and calculating PRS pleiotropy is available at https://github.com/andyw-dahl/mdd-impute and https://github.com/caina89/MDDImpute. The AutoComplete software is available at https://github.com/sriramlab/AutoComplete.

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

## Acknowledgements

A.D., S.-A.B., J.F., A.J.S., S.S., K.S.K. and N.C. are supported by R01MH130581 from the NIH. M.T. is supported in part by the NIH Training Grant in Genomic Analysis and Interpretation T32HG002536. U.A. and S.S. are supported in part by III-1705121, CAREER-1943497 and R35GM125055. V.A. is supported by Lundbeck Foundation postdoctoral grant R380-2021-1465. A.J.S. is supported by Lundbeckfonden Fellowship R335-2019-2318. R.B. is supported by T32-NS048004 from the NIH. The iPSYCH team was supported by grants from the Lundbeck Foundation (R102-A9118, R155-2014-1724 and R248-2017-2003), NIMH (1R01MH124851-01) and the Universities and University Hospitals of Aarhus and Copenhagen. The Danish National Biobank resource was supported by the Novo Nordisk Foundation. High-performance computer capacity for handling and statistical analysis of iPSYCH data on the GenomeDK HPC facility was provided by the Center for Genomics and Personalized Medicine and the Centre for Integrative Sequencing, iSEQ, Aarhus University, Denmark. We gratefully acknowledge the support of all collaborators and participants in UK Biobank, iPSYCH, CONVERGE and UCLA ATLAS who made this work possible.

## Author contributions

A.D. and N.C. wrote the paper. A.D., J.F., K.S.K. and N.C. designed the study. M.T. performed ATLAS analyses. U.A. and S.S. performed AutoComplete imputation. A.D. and N.C. performed all other analyses. M.K., V.A., T.W. and A.J.S. supported iPSYCH analyses. R.B. and S.-A.B. edited the manuscript. All authors reviewed the paper.

## Funding

## Competing interests

The authors declare no competing interests.

## Additional information

**Extended data** is available for this paper at https://doi.org/10.1038/s41588-023-01559-9.

**Correspondence and requests for materials** should be addressed to Andrew Dahl or Na Cai.

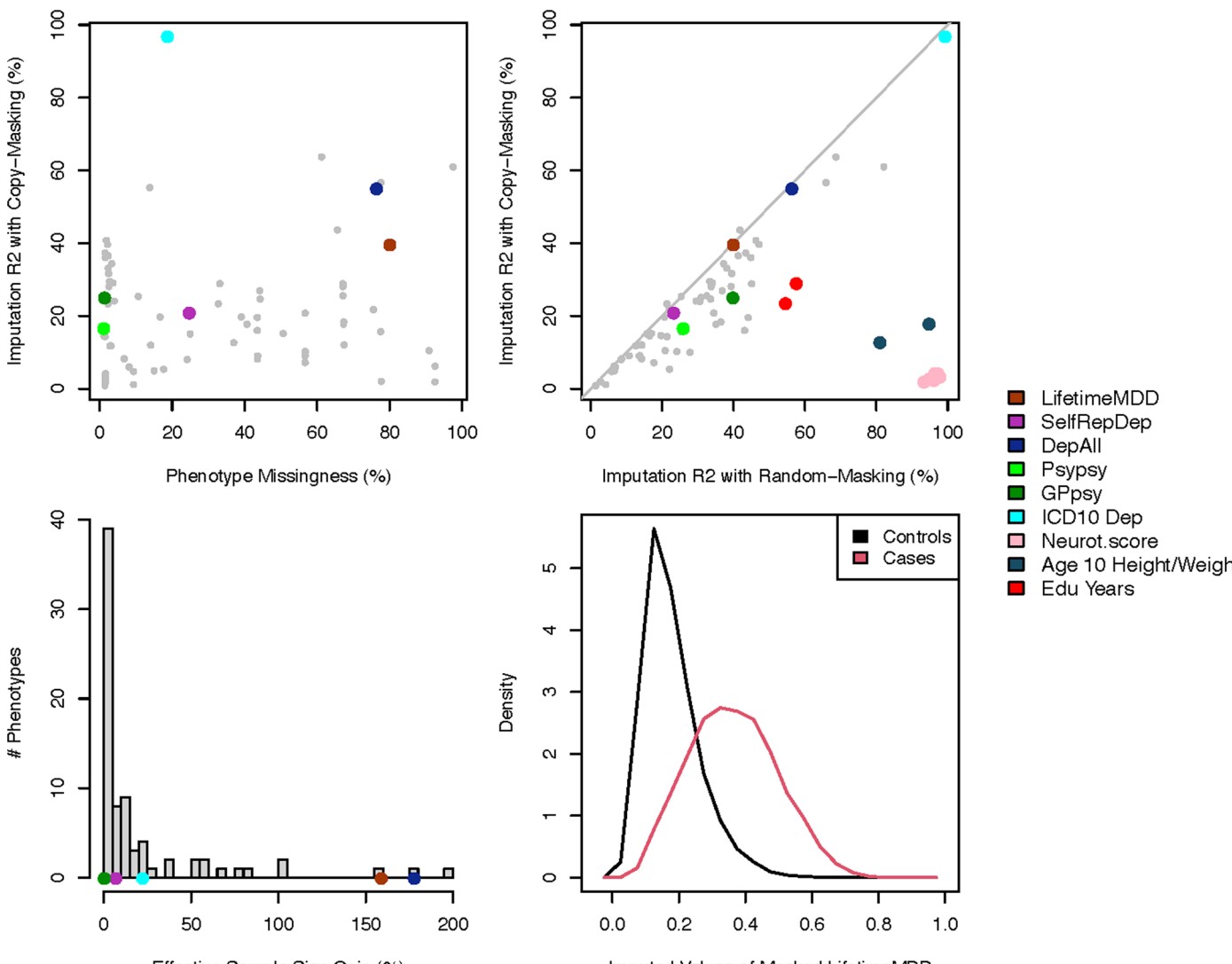

**Extended Data Fig. 1 | Imputation accuracy metrics across our depression-relevant UKB phenome. a**, Scatter plot of estimated imputation accuracy against phenotype missingness. **b**, Scatter plot of estimated imputation accuracy using our copy-masking approach against naive estimates masking entries uniformly at random. **c**, Distribution across phenotypes of gained effective sample size from phenotype imputation. **d**, Distribution of imputed LifetimeMDD values for held-out observations, which informally reflect the probabilities of having LifetimeMDD.

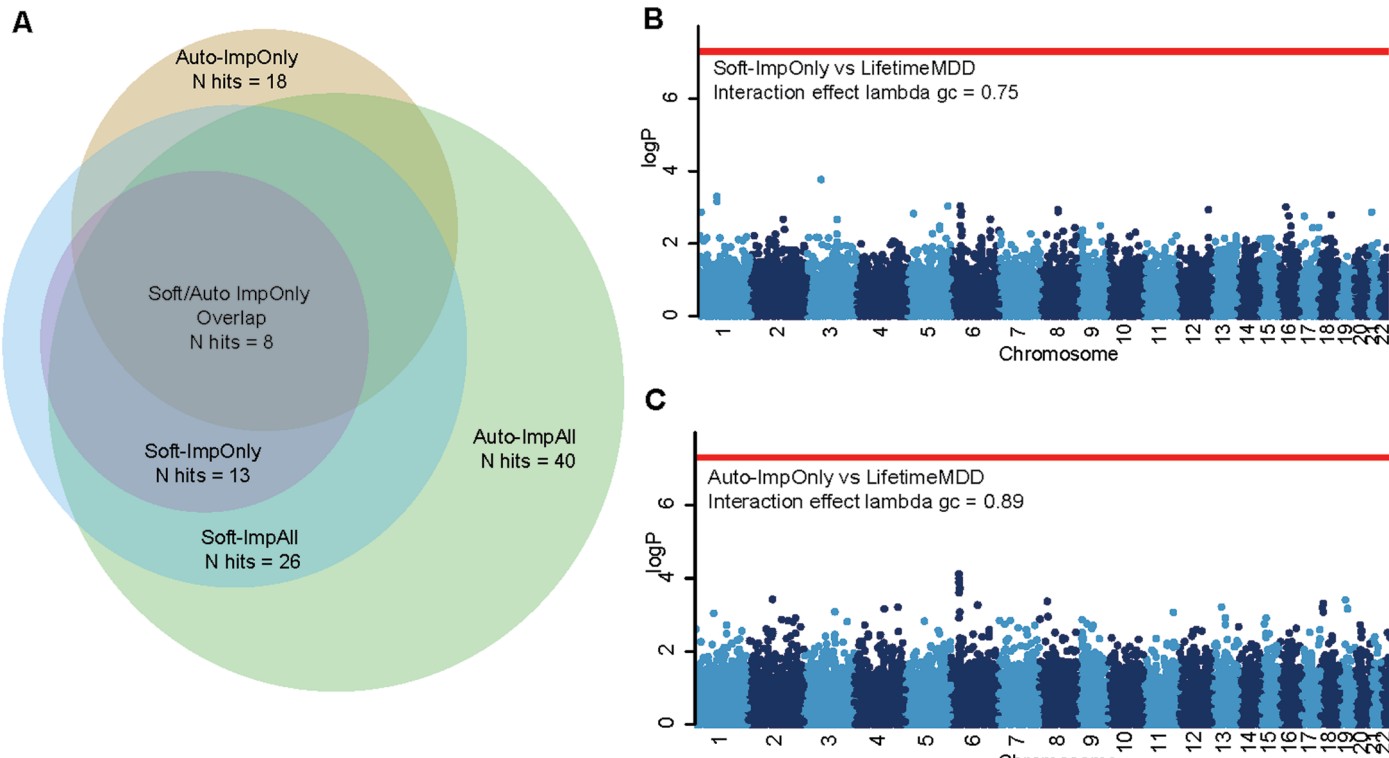

**Extended Data Fig. 2 | Quality control on GWAS loci identified from imputed LifetimeMDD. a**, Venn diagram showing the overlap of GWAS loci identified from GWAS on ImpOnly and ImpAll measures of LifetimeMDD from Softimpute and Autocomplete. **b,c**, Manhattan plots of Cochran's Q statistic's P-value for heterogeneity, obtained through a random effect meta-analysis performed with METASOFT, between genetic effects identified from GWAS on observed LifetimeMDD and GWAS on ImpOnly measures of LifetimeMDD; $-\log_{10}(P)$ values shown on the y-axis were before adjustment of multiple-testing; red line shows the genome-wide significance threshold corresponding to $P = 5 \times 10^{-8}$.

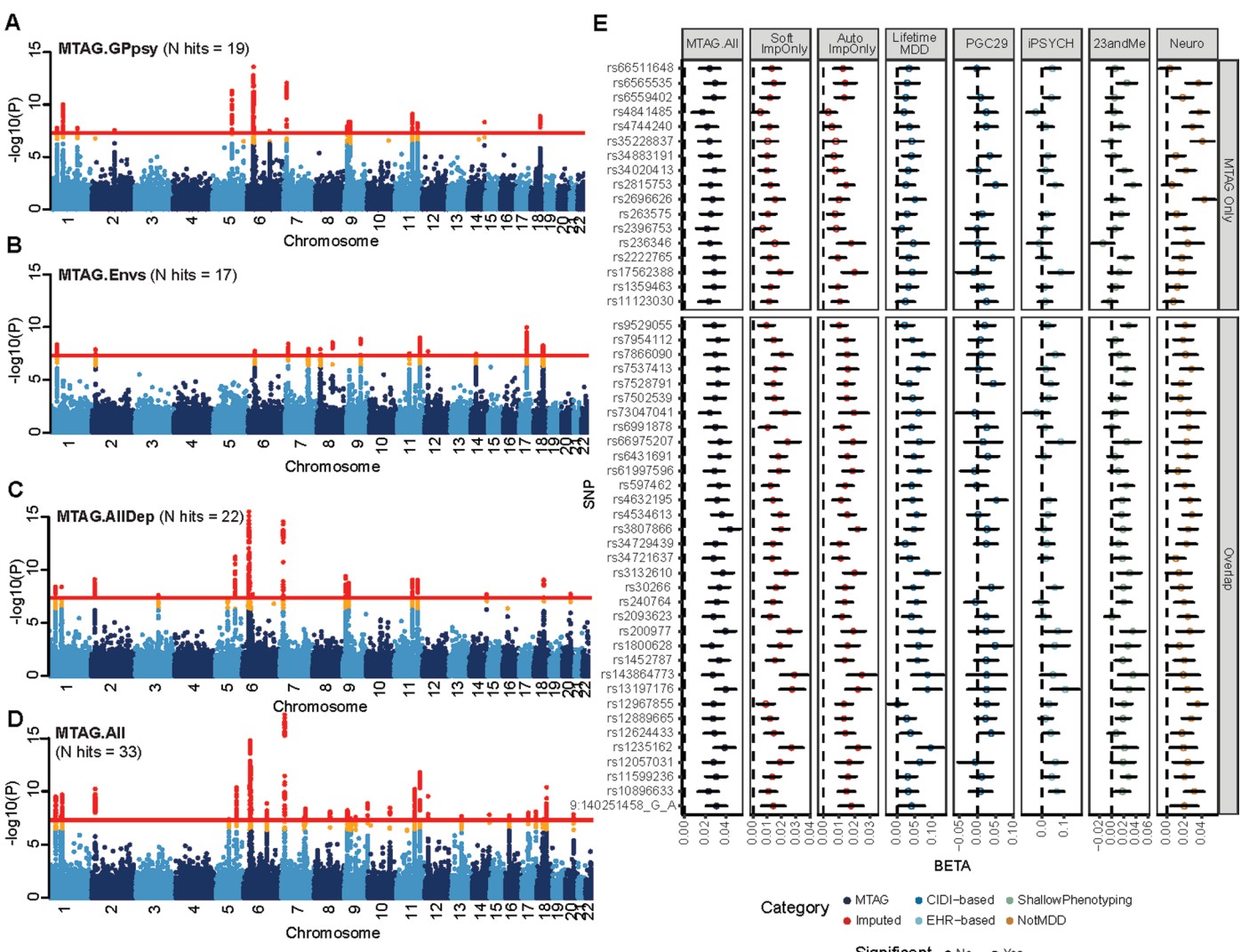

**Extended Data Fig. 3 | MTAG GWAS and genetic architecture. a-d,** Manhattan plots showing MTAG results for LifetimeMDD for the MTAG runs, MTAG.GPpsy, MTAG.Envs, MTAG.AllDep and MTAG.All, which are described in Fig. 4a; red line shows the genome-wide significance threshold ($P = 5 \times 10^{-8}$). **e,** Replication of LifetimeMDD GWAS effect sizes for loci identified only by MTAG and those identified by both MTAG and imputation (Softimpute or Autocomplete). Effect sizes are shown for observed LifetimeMDD and external MDD studies from PGC ($n = 42,455$), iPSYCH ($n = 38,128$), and 23andMe ($n = 307,354$). All error bars indicate 95% confidence intervals.

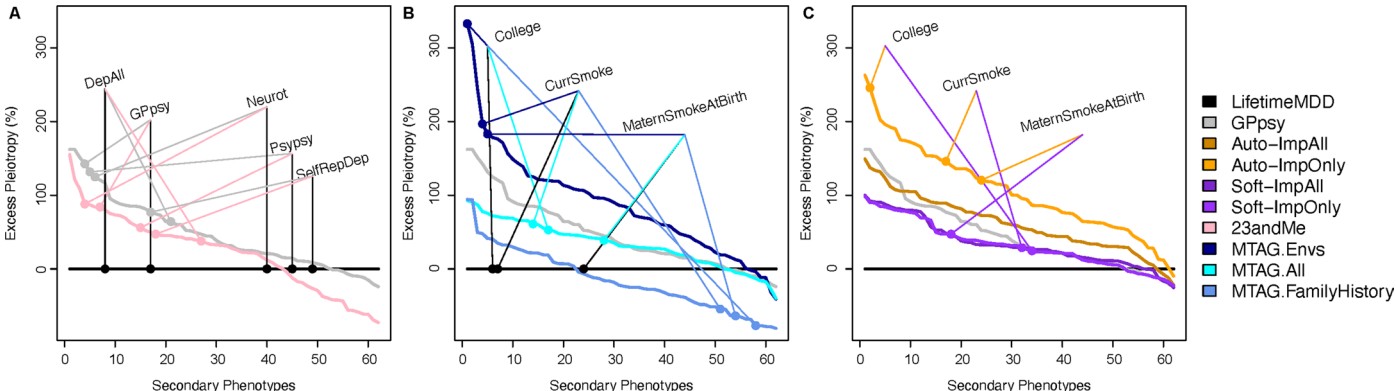

**Extended Data Fig. 4 | Excess PRS Pleiotropy.** Excess PRS Pleiotropy measures the pleiotropy of a given PRS relative to the PRS from observed LifetimeMDD. PRS Pleiotropy is defined as the PRS prediction ratio for a secondary trait relative to observed LifetimeMDD (PRS Pleiotropy = $R2_{secondary}/R2_{LifetimeMDD}$), and excess pleiotropy is the increase relative to the LifetimeMDD PRS (Excess PRS Pleiotropy = (PRS Pleiotropy − LifetimeMDD PRS Pleiotropy)/LifetimeMDD PRS Pleiotropy). Plots are ordered by Excess PRS Pleiotropy for each PRS. **a**, The PRS derived from GPpsy and 23andMe are less specific to LifetimeMDD than the observed LifetimeMDD PRS, especially for shallow MDD phenotypes and neuroticism. **b**, MTAG.Envs has high Excess PRS Pleiotropy to secondary traits like college education, smoking, and maternal smoking, while MTAG.FamilyHistory actually reduces PRS Pleiotropy for these traits. **c**, Both ImpOnly and ImpAll SoftImpute phenotypes show lower Excess PRS Pleiotropy than GPpsy, and the ImpAll GWAS from Autocomplete is comparable to GPpsy.

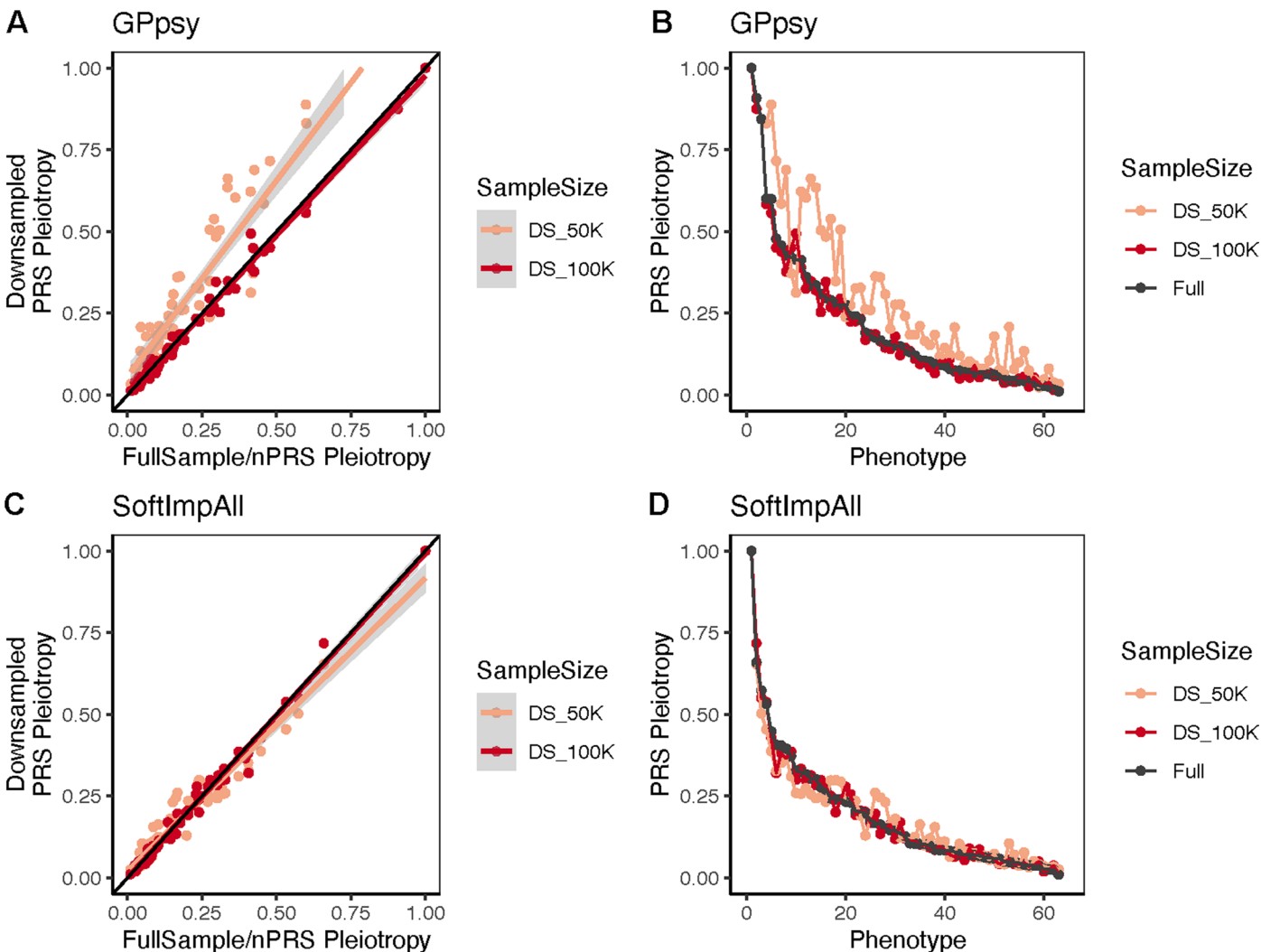

**Extended Data Fig. 5 | PRS Pleiotropy is robust to GWAS sample size in practice. a-d**, PRS Pleiotropy of GPpsy ($n = 332,629$) (**a,b**) and Soft-ImpAll ($n = 337,126$) (**c,d**) at full sample size and down-sampled to $n = 50K$ and 100K for 62 phenotypes in UKB. Plotted values show the mean PRS Pleiotropy from 10-fold cross-validation in UKB.

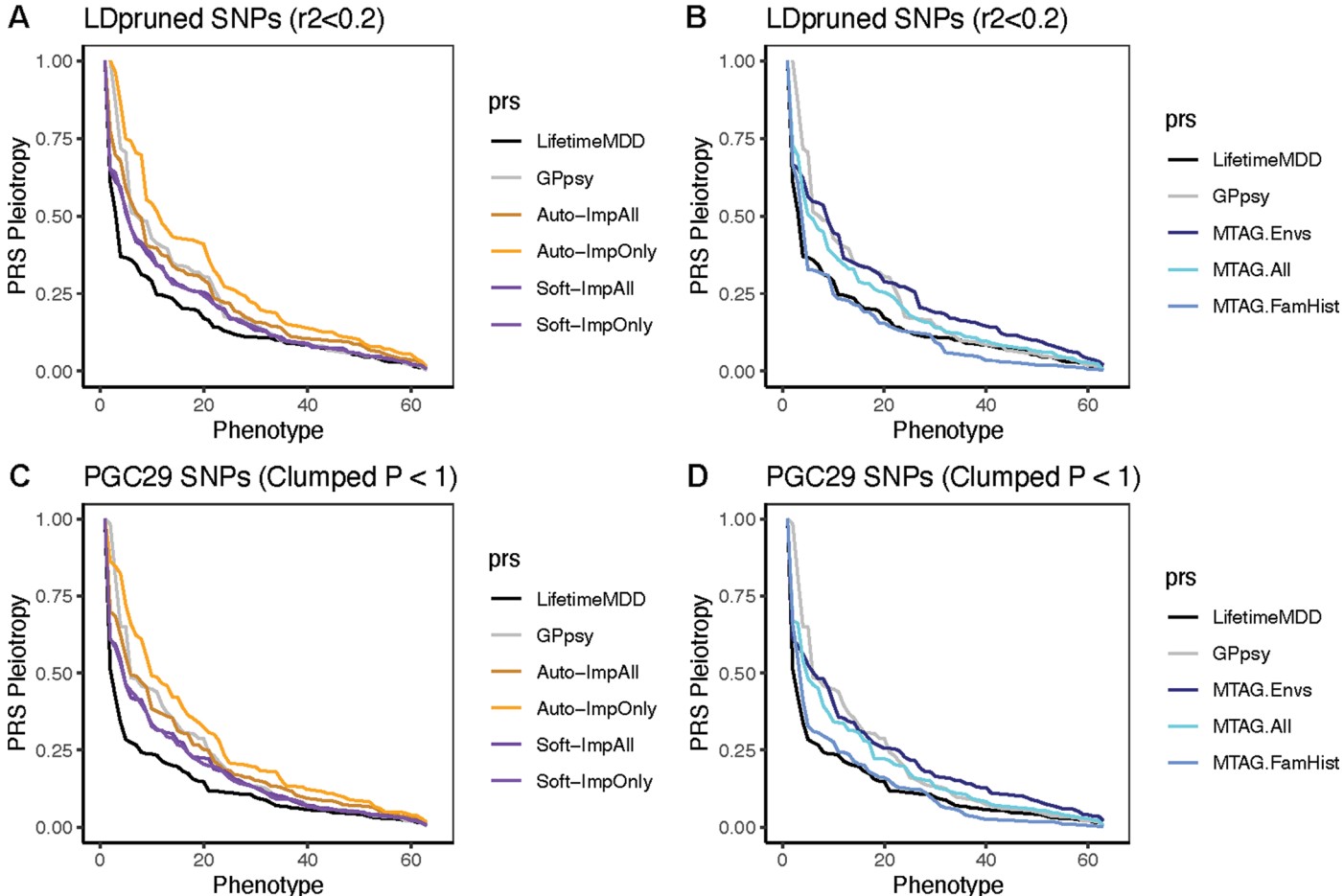

**Extended Data Fig. 6 | PRS Pleiotropy is robust to SNP pruning. a-d**, PRS Pleiotropy of imputed and MTAG GWAS for 62 phenotypes in UKB (which are significantly predicted by at least one full-sample PRS, as shown in Fig. 6), where the PRS is constructed with 136,563 LD-pruned SNPs ($r^2 < 0.2$) in UKB (**a,b**) and 91,315 SNPs from PGC29 GWAS clumped at $P_{threshold} = 1$ (**c,d**). Plotted values show the mean PRS Pleiotropy from 10-fold cross-validation in UKB.

# Reporting Summary

## Statistics

For all statistical analyses, confirm that the following items are present in the figure legend, table legend, main text, or Methods section.

| n/a | Confirmed | |
|---|---|---|
| ☐ | ☒ | The exact sample size (*n*) for each experimental group/condition, given as a discrete number and unit of measurement |
| ☐ | ☒ | A statement on whether measurements were taken from distinct samples or whether the same sample was measured repeatedly |
| ☐ | ☒ | The statistical test(s) used AND whether they are one- or two-sided<br>*Only common tests should be described solely by name; describe more complex techniques in the Methods section.* |
| ☐ | ☒ | A description of all covariates tested |
| ☐ | ☒ | A description of any assumptions or corrections, such as tests of normality and adjustment for multiple comparisons |
| ☐ | ☒ | A full description of the statistical parameters including central tendency (e.g. means) or other basic estimates (e.g. regression coefficient) AND variation (e.g. standard deviation) or associated estimates of uncertainty (e.g. confidence intervals) |
| ☐ | ☒ | For null hypothesis testing, the test statistic (e.g. *F*, *t*, *r*) with confidence intervals, effect sizes, degrees of freedom and *P* value noted<br>*Give P values as exact values whenever suitable.* |
| ☒ | ☐ | For Bayesian analysis, information on the choice of priors and Markov chain Monte Carlo settings |
| ☒ | ☐ | For hierarchical and complex designs, identification of the appropriate level for tests and full reporting of outcomes |
| ☐ | ☒ | Estimates of effect sizes (e.g. Cohen's *d*, Pearson's *r*), indicating how they were calculated |

*Our web collection on statistics for biologists contains articles on many of the points above.*

## Software and code

Policy information about availability of computer code

| Data collection | No new data was collected for this study. |
|---|---|
| Data analysis | We conducted our analyses using the following published and publicly available software: 1) for computing PCA on all White British sample in UKBiobank: flashPCA v2.0 (https://github.com/gabraham/flashpca); 2) for GWAS using logistic regression: PLINK v2 (https://www.cog-genomics.org/plink2); 3) for estimation of heritability, genetic correlation and/or enrichment of heritability: LDSC v1.0.1 (https://github.com/bulik/ldsc); 4) for calculating polygenic risk scores: PRSice v2 (https://www.prsice.info/); 5) for imputing phenotypes in the MDD-related phenome: softImpute v1.4-1 (https://cran.r-project.org/web/packages/softImpute/index.html); 6) for imputing phenotypes in the MDD-related phenome with AutoComplete: https://github.com/sriramlab/AutoComplete; 7) for phenotype integration at the summary statistics level: MTAG v0.9.0 (https://github.com/JonJala/mtag); 8) for testing heterogeneity of effects between non-overlapping GWAS: METASOFT (http://genetics.cs.ucla.edu/meta/). We further provide custom code for running softImpute and calculating PRS Pleiotropy in https://github.com/andywdahl/mdd-impute and https://github.com/caina89/MDDImpute; this is reported in our Code Availability statement. |

For manuscripts utilizing custom algorithms or software that are central to the research but not yet described in published literature, software must be made available to editors and reviewers. We strongly encourage code deposition in a community repository (e.g. GitHub). See the Nature Portfolio guidelines for submitting code & software for further information.

## Data

Policy information about availability of data

All manuscripts must include a data availability statement. This statement should provide the following information, where applicable:
- Accession codes, unique identifiers, or web links for publicly available datasets
- A description of any restrictions on data availability
- For clinical datasets or third party data, please ensure that the statement adheres to our policy

We used the genotype and phenotype data from A) 502,637 samples in the full release (imputation version 2) of the UK Biobank Resource under application no. 28709 and 33217; B) iPSYCH cohorts 2012 and 2015i with genotype data and phenotype data on Major Depressive Disorder (2012: Ncontrols = 23,371, Ncases = 18,879; 2015i: Ncontrols = 15,163, Ncases = 8,188; Total: Ncontrols = 38,534, Ncases = 27,067); C) UCLA ATLAS electronic health record cohort where genotype data and phecode for depressive disorders and the subset Major Depressive Disorder were available (1,997 unrelated Asian-identifying individuals, 1,125 unrelated Black-identifying individuals, 2,169 unrelated Latino-identifying individuals, and 14,366 unrelated White-identifying individuals, see Supplemental Tables 5-6). Details of all cohorts used and both genotype quality control and phenotype selection are described in Supplementary Note. We further used publicly available summary statistics from other studies downloadable from the website of Psychiatric Genomics Consortium (https://www.med.unc.edu/pgc/results-and-downloads) and figshare for iPSYCH2012 (https://doi.org/10.6084/m9.figshare.20517330), and the references for which can be found in Supplemental Table 4. The individual-level CONVERGE, Danish and UCLA datasets are not publicly available due to institutional restrictions on data sharing and privacy concerns. We provide summary statistics of all GWAS described in this study on https://doi.org/10.6084/m9.figshare.19604335.

## Human research participants

Policy information about studies involving human research participants and Sex and Gender in Research.

| | |
|---|---|
| Reporting on sex and gender | We included self-reported sex/gender as a phenotype in our MDD-relevant phenome and the analysis of the latent factors from softImpute showed that sex/gender is reflected by one of the top factors accounting for variation in the MDD-related phenome (factor 5). We explain this in the manuscript. We further investigated if results of phenotype imputation differs if we stratified the data by sex and performed imputation separately in each sex; we found that the results of the sex-stratified imputation correlate highly with the joint-imputation, and for most phenotypes, including our focal phenotype LifetimeMDD, imputation does better when performed jointly on both sexes. |
| Population characteristics | We clearly describe characteristics of each studied cohort in our Methods and Supplementary Materials. In brief: (1) UK Biobank contains older British individuals (age range 37-73 at point of data collection), and we stratify our analyses using a combination of self-reported ethnicity and genetically-informed continental-level ancestry; (2) ATLAS contains diverse individuals in the UCLA medical system (age 18-86 at point of data collection); (3) iPSYCH contains Danish individuals who are drawn randomly from the population or have diagnoses of common mental disorders (age range 8-32 for iPSYCH2012, age range 8-35 for iPSYCH2015i, all ages at point of data collection); (4) CONVERGE contains Chinese women screened by mental health professionals (age range 30-60 at point of data collection). |
| Recruitment | No new data was collected for this study. |
| Ethics oversight | This research was conducted under the ethical approval from the UK Biobank Resource under application no. 28709 and 33217. The use of iPSYCH data follows standards of the Danish Scientific Ethics Committee, the Danish Health Data Authority, the Danish Data Protection Agency, and the Danish Neonatal Screening Biobank Steering Committee. Data access was via secure portals in accordance with Danish data protection guidelines set by the Danish Data Protection Agency, the Danish Health Data Authority, and Statistics Denmark. Retrospective data collection and analysis for ATLAS was approved by the UCLA IRB39. Patient Recruitment and Sample Collection for Precision Health Activities at UCLA is an approved study by the UCLA Institutional Review Board (UCLA IRB17-001013). All necessary patient/participant consent has been obtained and the appropriate institutional forms have been archived. The CONVERGE (China, Oxford, and VCU Experimental Research on Genetic Epidemiology) study was approved by the ethical review boards of Oxford University and participating hospitals. All participants provided written informed consent. |

Note that full information on the approval of the study protocol must also be provided in the manuscript.

# Field-specific reporting

Please select the one below that is the best fit for your research. If you are not sure, read the appropriate sections before making your selection.

☒ Life sciences ☐ Behavioural & social sciences ☐ Ecological, evolutionary & environmental sciences

For a reference copy of the document with all sections, see nature.com/documents/nr-reporting-summary-flat.pdf

# Life sciences study design

All studies must disclose on these points even when the disclosure is negative.

| | |
|---|---|
| Sample size | We used the genotype and phenotype data from A) 502,637 samples in the full release (imputation version 2) of the UK Biobank Resource |

| Sample size | under application no. 28709 and 33217; B) iPSYCH cohorts 2012 and 2015i with genotype data and phenotype data on Major Depressive Disorder (2012: Ncontrols = 23,371, Ncases = 18,879; 2015i: Ncontrols = 15,163, Ncases = 8,188; Total: Ncontrols = 38,534, Ncases = 27,067); C) UCLA ATLAS electronic health record cohort where genotype data and phecode for depressive disorders and the subset Major Depressive Disorder were available (1,997 unrelated Asian-identifying individuals, 1,125 unrelated Black-identifying individuals, 2,169 unrelated Latino-identifying individuals, and 14,366 unrelated White-identifying individuals, see Supplemental Tables 7-8). For all datasets, sample sizes were determined by the maximum number of individuals collected by the cohort with information on depression-related phenotypes rather than power analyses. Much of the work in this manuscript explores how much power these sample sizes have in GWAS and PRS analyses. |
|---|---|
| Data exclusions | We excluded all samples with 1) poor genotyping quality, 2) high level of relatedness to other samples, 3) ancestries other than White British as indicated by the QC metrics from UKBiobank (Bycroft et al 2018, https://doi.org/10.1038/s41586-018-0579-z), 4) sex chromosome aneuploidy, 5) withdrawal of consent from being included in research on data from the UKBiobank, 6) a history of substance abuse, and 7) manic or psychotic conditions. This gives us our final sample of 337,198 White-British, unrelated individuals. Details of exclusion criteria can be found in Supplemental Methods section "Sample filtering". |
| Replication | We replicated significant genetic effects identified in GWAS on imputed LifetimeMDD in UKBiobank (with both softImpute and Autocomplete) using summary statistics from external cohorts of MDD: PGC29, 23andMe and iPSYCH (this is described in Methods and references of the cohorts used can be found in Supplementary Table 4). We then replicated the improvement in PRS prediction accuracy in imputed and MTAG PRS, both in individuals of European ancestry and in other ancestries, in iPSYCH, ATLAS and CONVERGE. |
| Randomization | Not applicable, as no new assessments were performed in this study and all analyses were conducted on data from all individuals present in existing data. |
| Blinding | Not applicable, as data was collected by the time this study is conducted, all assessments on participants of cohorts used in this analyses were not blinded (if clinicians were involved, in CONVERGE, iPSYCH, UCLA ATLAS, and for all ICD codes in UKBiobank) or self-administered (for all self-reported depression related phenotypes and depression measures in UKBiobank). |

# Reporting for specific materials, systems and methods

We require information from authors about some types of materials, experimental systems and methods used in many studies. Here, indicate whether each material, system or method listed is relevant to your study. If you are not sure if a list item applies to your research, read the appropriate section before selecting a response.

## Materials & experimental systems

| n/a | Involved in the study |
|---|---|
| ☒ | ☐ Antibodies |
| ☒ | ☐ Eukaryotic cell lines |
| ☒ | ☐ Palaeontology and archaeology |
| ☒ | ☐ Animals and other organisms |
| ☒ | ☐ Clinical data |
| ☒ | ☐ Dual use research of concern |

## Methods

| n/a | Involved in the study |
|---|---|
| ☒ | ☐ ChIP-seq |
| ☒ | ☐ Flow cytometry |
| ☒ | ☐ MRI-based neuroimaging |

