## [Peer Review File · Nature Genetics]

Peer Review Information

Manuscript Title: Phenotype integration improves power and preserves specificity in biobank-based genetic studies of Major Depressive Disorder

Corresponding author name(s): Dr Na Cai, Dr Andrew Dahl

Reviewer Comments & Decisions:

Decision Letter, initial version:

21st October 2022

Dear Na,

Your Article "Phenotype integration improves power and preserves specificity in biobank-based genetic studies of MDD" has been seen by three referees. You will see from their comments below that, while they find your work of interest, they have raised several important points. We are interested in the possibility of publishing your study in Nature Genetics, but we would like to consider your response to these points in the form of a revised manuscript before we make a final decision on publication.

To guide the scope of the revisions, the editors discuss the referee reports in detail within the team, including with the chief editor, with a view to identifying key priorities that should be addressed in revision and sometimes overruling referee requests that are deemed beyond the scope of the current study. In this case, we ask that you carefully address all technical queries related to the analyses and their interpretation, placing the work in context with prior studies on phenotype imputation and extending the analyses where feasible as suggested by the referees. We hope you will find this prioritized set of referee points to be useful when revising your study. Please do not hesitate to get in touch if you would like to discuss these issues further.

We therefore invite you to revise your manuscript taking into account all reviewer and editor comments. Please highlight all changes in the manuscript text file. At this stage, we will need you to upload a copy of the manuscript in MS Word .docx or similar editable format.

*1) Include a "Response to referees" document detailing, point-by-point, how you addressed each

referee comment. If no action was taken to address a point, you must provide a compelling argument. This response will be sent back to the referees along with the revised manuscript.

*2) If you have not done so already please begin to revise your manuscript so that it conforms to our Article format instructions, available [here](http://www.nature.com/ng/authors/article_types/index.html). Refer also to any guidelines provided in this letter.

[redacted]

We hope to receive your revised manuscript within 8-12 weeks. If you cannot send it within this time, please let us know.

Sincerely,
Kyle

Kyle Vogan, PhD
Senior Editor
Nature Genetics
<https://orcid.org/0000-0001-9565-9665>

Referee expertise:

Referee #1: Genetics, psychiatric disorders, statistical methods

Referee #2: Genetics, psychiatric disorders, major depression

Referee #3: Genetics, complex traits, statistical methods

Reviewers' Comments:

Reviewer #1:

Remarks to the Author:

Dahl et al provide a novel analysis investigating analysis of the UKB investigating the validity of phenotype imputation. This study is important for biobanks or other cohorts particularly longitudinal cohorts where data subsets have deep phenotyping while the balance of data have shallow phenotyping. It is a very nice study and clearly signposted throughout.

Major Points

1) Motivation/importance of the results. The study addresses the question if phenotype imputation – making estimated missing phenotypes for individuals - improves upon meta-analysis of correlated traits in genetic studies. However the motivation of the study could be given a different emphasis as I think there may be reasons why it is useful to have non-missing individual-level data in biobank data, particularly when there is longitudinal. For example, measures at baseline could be useful as predictors of longitudinal phenotypic outcome. With this motivation, the genetics analyses provide support for the validity of the imputation. At the moment, despite the talking up of the phenotype imputation, if the goal is GWAS discovery and PRS then MTAG seems superior. MTAG outperforms phenotype imputation in R2 and PRS sensitivity or specificity. This suggestion implies a rewrite of the Discussion.

2) Number of traits.

a. The imputation uses 216 phenotypes. Supp Table 1 has 247 rows. It would be helpful if these number matched or if a column was labelled to indicate in/out of 216. Also columns to show which variables were used in the MTAG options.

b. The first reason for not using 216 phenotypes in MTAG seems unreasonable, given that many groups have provided GWAS results for all UKB traits, and GWAS and LDSC etc runs are very fast, and should be deleted.

c. The second reason for not using 216 phenotypes in MTAG “MTAG accrues false positives as the number of traits grows” makes sense. In fact, so much so that one wonders if including 216 traits in the trait imputation may introduce more noise than if the number of traits were limited. It would be useful to benchmark MTAG.All by imputation that uses the same traits (or same traits + sex + age plus demographic traits).

3) Sex is used as a variable in the imputation, but it seems likely that the imputation could be

improved by conducting it within sex?

4) It seems strange that BMI was not included as a variable in the imputation. Also strange not considered in the pleiotropy analyses?

5) Ideally the PGC29 cohorts could have been considered in Fig 5B as they are LifetimeMDD, but that is likely a big job. The UKB GWAS sumstats from the phenotype imputation and MTAG need to be made available to allow other to do this.

Minor comments:

1. The term "effective" sample size usually refers to a specific power-related statement. For example, the effective sample size of a case control cohort is the power of the equivalent power of the actual cohort $N_{case} < N_{control}$ expressed as the power of a sample size where $N_{case} = N_{control}$. Here, although the power is certainly increased as a reflection of sample size, and the power doesn't reflect the actual sample size because essentially a correlated trait is measured, but the power is not quantified. So the use of "effective" should be avoided. Or is this the effective N measured from MTAG? In which case effective it is OK.

2. Suggest that the PRS in non-EUR is placed in supplement as it seems not very interpretable, but good to have tried.

3. This section "Phenotype integration improves PRS portability" should be "Phenotype integration and MTAG improve PRS portability".

4. SNP-based heritability is now the preferred term as SNP-heritability is ambiguous.

5. When introducing the cohorts 23andMe, PGC29 etc in the written it would be good to add effective N (as in usual definition!) as a quick benchmark.

6. More detail is needed for the estimates of SNP-based heritability. Make a table with N is used in LD score regression and the K factor for conversion to liability and outcomes. These data will help other researchers accessing the sumstats.

7. Fig 6. Use wording in legend to make it clear that the LifetimeMDD and GPsy are the same in all figures.

8. Extended Data Fig 1. Since LifetimeMDD is the key trait, can you add a 2x2 table of Measured/predicted data that generates the brown dot in part a?

Reviewer #2:

Remarks to the Author:

This is an interesting study that uses phenotype imputation to derive MDD-like traits that are similar in their associations to a CIDI-derived MDD diagnosis. The findings could lead to larger sample sizes from UK Biobank and other studies and might lead to results that are more relevant to MDD research, than the broad phenotypes currently widely-used.

The work is somewhat novel in MDD, although there are other examples of phenotype imputation,

such as <https://www.biorxiv.org/content/10.1101/2022.08.15.503991v1.abstract> (using a more complex approach) and Glanville (<https://www.cambridge.org/core/journals/bjpsych-open/article/multiple-measures-of-depression-to-enhance-validity-of-major-depressive-disorder-in-the-uk/D34B7DCDD744B0C3520C7AEAE6A8E718>). The results aren't entirely novel, but takes a different methodological approach to these studies.

The authors state that MDD's treatments are 'relatively ineffective' but make no comparison. MDD treatments compare well in terms of NNT to many for physical disorders and are similarly effective to many psychological and physical treatments for other psychiatric disorders. I think this statement is oversimplified and misleading and should either be made more explicit (with % treatment failures), compared to treatments with other disorders, or removed.

The shallow-deep phenotype divide is framed in a way that can't currently be confirmed or supported using the studies they cite. They state that deeply-phenotyped studies have produced smaller numbers of loci that could generate hypotheses on MDD biology. However, that does not stand up to scrutiny – the studies they cite are volunteer studies and volunteers in studies of psychiatric disorders are frequently better educated and resourced than the population at large. Biological inferences from these studies are highly problematic when there are very few (sometimes zero) replicated loci – as those inferences also depend on the number of significant loci and genetic signals that can be partitioned into pathways and processes (for example).

Health record measures of MDD are also grouped with other broad phenotypes, although these are perhaps the most representative of the population as a whole and most likely to resemble those who would gain from insights and treatments developed on the back of GWAS. The iPSYCH study mentioned is also one whose methods are based partly on Danish electronic health records.

The statement that broad phenotypes generate signals that are not specific for MDD is also questionable when those samples lack power, and an equal comparison (without down-sampling broad phenotyping studies) is problematic as the ground truth is not known.

This manuscript essentially reports missing phenotype imputation. There is an enormous literature on this topic, and discussion of the pitfalls and limitations. Reference to this literature is largely missing from the paper.

The CIDI-SF is assumed to be the preferred phenotype in UKB. People who completed this questionnaire are atypical of the whole in several respects (Adams M, 2018 IJE: <https://academic.oup.com/ije/article/49/2/410/5526901>). On the back of UKB, whose participation rate was ~6%, this sample is clearly very highly selected. Have the authors considered the biases that may result from this selection?

One of the many limitations of missing value imputation is that it reduces the variance of the phenotype of interest which, if observed, would deviate randomly from the proposed model used to derive missing values. Can the authors clarify whether this was something they saw in the imputed data and whether they think it's an important consideration?

The MDD-related traits used for imputation are the ones correlated with MDD in the training sample. These are presumably considered to be those that are relevant by the authors, but it's clear that in the imputed samples those variables will also be correlated with MDD, maybe more so than if they had been measured directly. Do the authors think this is a problem? Do the authors think that their analyses could be accused of being circular – that pattern of genetic correlations used to support their methods are influenced by the traits used for imputation?

The increased number of GWAS loci identified in the observed vs imputed MDD GWAS analysis is not a good indication of power when the number of gwas loci is low. Repeated GWAS analyses of the same phenotype in samples of equal size can generate different numbers of loci by the play of chance alone. Do the authors think there is a better approach to this question? If they took effect sizes from another independent GWAS-MA study, would that help to identify whether they had more power to discover significant signals using a fairer test?

Why were heritability estimates on p_5 provided on the observed rather than on the liability scale?

The authors quote a paper by Glanville et al that used a different approach to missing values – creating a composite phenotype that also enlarged sample size. There aren't any direct comparisons with this approach. I think those would be interesting and useful.

Reviewer #3:

Remarks to the Author:

This is a fantastic paper that is well written and explores important empirical and methodological questions. In this paper, the authors discuss the trade-off between power and specificity. For many phenotypes, researchers must choose between small samples that are well phenotyped and large samples with less precise measurement. Poorly measured phenotypes can lead to reduced power (at best) and systematic biases in GWAS results (at worst). Similarly, methods that claim to increase power (such as MTAG) may also induce these sorts of biases. The authors use the application of major depressive disorder (MDD) to assess how these concerns play out in that setting. They propose a series of imputation approaches that can boost power without large risks of contamination and loss of specificity.

Overall, I think the paper is in great shape. I have a small number of recommendations that the authors may consider integrating into a revision to clarify some of their analyses and connect it to existing work.

1) A very similar question is explored theoretically in Okbay et al. (2016). (See Supplementary Note Section 3.) How does the authors' analysis compare to the work done in that paper?

2) Hill et al. (2019) also noted that MTAG-based summary statistics have greater power but seem to be less specific. For example, the genetic correlation between intelligence and educational attainment is 0.7 when using the raw summary statistics, but it's $>.8$ when using MTAG-based summary statistics for intelligence that leverage the educational attainment summary statistics. While this paper is a much more thorough treatment of that issue, I believe that this is the same phenomenon that the authors of this paper find when they discuss specificity. Do the authors agree or is there something unique about MDD?

3) I am having a hard time interpreting the PRS pleiotropy measure proposed by the authors. First, if the baseline R^2 for LifetimeMDD is different for the different PRSs, then how comparable is the ratio across PRSs. Given that R^2 is nonlinear in the effective sample size of the underlying GWAS, would we expect the PRS pleiotropy to be the same if we increased the sample size of the deep MDD phenotypes such that the baseline R^2 was the same? Relatedly, the authors use PRSice to construct their PRSs in this paper. This approach only uses a subset of SNPs that meet some p -value threshold. As a result, GWASs with greater power will use more SNPs that have smaller associations with MDD

and which may be more pleiotropic in general. How does the PRS Pleiotropy compare when using the same set of SNPs in the various PRSs but using the different sets of GWAS estimates considered?

4) On page 8, the authors claim "If phenotype integration captures core MDD biology, it will improve PRS predictions regardless of the context". This also seems a bit too strong. Why can't core biology operate through context dependent pathways? For example, the main SNP on CHRNA5 that predicts smoking behavior will operate more strongly in communities where smoking is more prevalent. You could imagine genes that play a core biological function by affecting the way the body responds to environmental factors that vary by context. The only way I can think that the above statement is necessarily true is if "core MDD biology" is defined as factors that are independent of context, in which case the statement is somewhat tautological.

5) Typo: I think there may be a missing word on page 9, in section titled "Phenotype integration improves PRS portability." It currently says, "Finally, we also for PRS prediction accuracy in UKB individuals with Asian ancestry..."

References:

Okbay, A., Baselmans, B. M., De Neve, J. E., Turley, P., Nivard, M. G., Fontana, M. A., ... & Rich, S. S. (2016). Genetic variants associated with subjective well-being, depressive symptoms, and neuroticism identified through genome-wide analyses. *Nature genetics*, 48(6), 624-633.

Hill, W. D., Marioni, R. E., Maghzian, O., Ritchie, S. J., Hagenaars, S. P., McIntosh, A. M., ... & Deary, I. J. (2019). A combined analysis of genetically correlated traits identifies 187 loci and a role for neurogenesis and myelination in intelligence. *Molecular psychiatry*, 24(2), 169-181.

Author Rebuttal to Initial comments

Referee expertise:

Referee #1: Genetics, psychiatric disorders, statistical methods

Referee #2: Genetics, psychiatric disorders, major depression

Referee #3: Genetics, complex traits, statistical methods

Reviewers' Comments:

Reviewer #1:

Remarks to the Author:

Dahl et al provide a novel analysis investigating analysis of the UKB investigating the validity of

phenotype imputation. This study is important for biobanks or other cohorts particularly longitudinal cohorts where data subsets have deep phenotyping while the balance of data have shallow phenotyping. It is a very nice study and clearly signposted throughout.

We thank the reviewer for their enthusiasm and constructive comments. Responding to these comments has significantly strengthened our empirical evaluation of phenotype imputation and improved the clarity of our paper in several important ways.

Major Points

1) Motivation/importance of the results. The study addresses the question if phenotype imputation – making estimated missing phenotypes for individuals - improves upon meta-analysis of correlated traits in genetic studies. However the motivation of the study could be given a different emphasis as I think there may be reasons why it is useful to have non-missing individual-level data in biobank data, particularly when there is longitudinal. For example, measures at baseline could be useful as predictors of longitudinal phenotypic outcome. With this motivation, the genetics analyses provide support for the validity of the imputation. At the moment, despite the talking up of the phenotype imputation, if the goal is GWAS discovery and PRS then MTAG seems superior. MTAG outperforms phenotype imputation in R2 and PRS sensitivity or specificity. This suggestion implies a rewrite of the Discussion.

i) To clarify our goals and nonstandard terminology, our paper is about phenotype integration, which we define to include both MTAG and phenotype imputation. Our idea for this term is to have a wide umbrella capturing a variety of approaches to combine together related measures, which are often found in biobank-based analyses of complex disorders—in other words, we would be happy to see approaches integrate together phenotypes that outperform both MTAG and imputation. We apologize for the confusing terminology, and have made several edits to improve clarity (please see Minor 3, below).

ii) Overall, it is not obvious to us that either MTAG or imputation is superior for GWAS/PRS. Instead, our main message is that phenotype integration (=imputation+MTAG) can significantly improve power and specificity in biobank-based MDD GWAS. We now make clear we mean both imputation and MTAG when we say “integration” in the first sentence of the Discussion:

In this paper, we address the power-specificity tradeoff between deep and shallow MDD phenotypes by integrating them together using phenotype imputation or MTAG

We also now emphasize this when concluding Discussion paragraph 1:

For MTAG, adding more phenotypes significantly increases power but can sacrifice specificity. Imputing LifetimeMDD with SoftImpute, on the other hand, preserves more specificity than shallow MDD phenotypes. Overall, our results demonstrate that both approaches to phenotype integration

outperform the observed deep or shallow phenotypes alone, and that phenotype integration is a practical way to improve biobank-based GWAS.

iii) MTAG performs great in general, and we characterize how its performance varies with input GWASes. MTAG.All is optimal in terms of #GWAS hits and PRS R2. MTAG.FamHist improves specificity, which we found surprising and exciting. We modified the Discussion to better emphasize these strengths of MTAG (please also see tracked changes).

MTAG has several important strengths and complements imputation. First, it operates at the summary statistics level, which enables incorporating external GWAS results and is far more computationally efficient than phenotype imputation once GWAS have been performed. Second, we found that MTAG.All generally outperforms imputation in terms of GWAS hits and PRS power and portability. However, the tradeoff is that MTAG generally has less specificity. A striking exception is MTAG.FamHist, which preserves specificity. This extends previous observations that careful methods can exploit family history to improve genetic studies^{43,44,55}.

As a related point, we restructured the Discussion to have a paragraph focused on phenotype imputation and another on MTAG, each discussing relative strengths, limitations, and future directions. We hope this helps clarify and emphasize the complementarity of these two frameworks for phenotype integration.

iv) But MTAG is less specific than imputation (except MTAG.FamHist, which has low power). Extended Data Figure 4 shows how MTAG amplifies MDD-nonspecific genetic signals. MTAG also inflates heritability and lambda_GC (Figure 4). MTAG also relies heavily on optimizing inputs, which is challenging and computationally expensive (please see Major 2c below).

v) The multi-trait factor analyses (Figure 3) require using softImpute instead of MTAG (though GenomicSEM may be an MTAG-like alternative, as we say in our Discussion).

v) The relevance to longitudinal data is a great point, which we have added to the Discussion:

A particularly important but challenging future application for phenotype imputation will be to longitudinal data, which is often sporadically measured across time points and individuals, especially in biobank data. This requires carefully modeling non-random missingness, which is expected to be severe in longitudinal data.

2) Number of traits.

a. The imputation uses 216 phenotypes. Supp Table 1 has 247 rows. It would be helpful if these number matched or if a column was labelled to indicate in/out of 216. Also columns to show which variables were used in the MTAG options.

i) 247 vs 216: Thank you for catching this. We apologize for causing confusion. 216 is the number of phenotypes we used in imputation. 247 is the number of lines in Supplementary Table 1. The mismatch is because phenotypes and Supplementary Table 1 lines are not one-to-one:

- A. 11 disease phenotypes each occupy 5-6 lines (lines 47 to 105) as they are based on a combination of items from the UKB data
- B. Two ICD10 code-based definitions of MDD (lines 242-254, 2 lines each)
- C. PCs1-20 occupy a single line (235) instead of 20 columns of the phenotype imputation matrix
- D. We added BMI to the table, which is used in a new secondary analysis (Major 4 below)

ii) In checking this table, we realized that we previously missed one phenotype (“educatedyrs”). We have now fixed this by adding the missing line and by updating the manuscript accordingly (the total number of phenotypes is 217, not including BMI; after including BMI, the total number of phenotypes is 218).

iii) To improve clarity, we have added the following line to the caption of Supplementary Table 1:

Note that phenotypes included in our imputation are not one-to-one with lines in the Table, because some lines correspond to multiple phenotypes (e.g., PC1-20) and some phenotypes correspond to multiple lines (e.g., ICD10 code-based definitions of MDD).

iv) We have added a column in Supplementary Table 1 indicating which variables were included in which MTAG runs.

b. The first reason for not using 216 phenotypes in MTAG seems unreasonable, given that many groups have provided GWAS results for all UKB traits, and GWAS and LDSC etc runs are very fast, and should be deleted.

i) We agree that computation is trivial when using existing GWAS results, and that this is important. We have added a comment to our Discussion to emphasize this:

MTAG has several important strengths and complements imputation. First, it operates at the summary statistics level, which enables incorporating external GWAS results and is far more computationally efficient than phenotype imputation once GWAS have been performed.

ii) However, our point is about optimizing the choice of MTAG inputs via cross-validation, which cannot use existing GWAS summary statistics. Cross-validation is essential for our work. First, MTAG depends heavily on the inputs, which we unbiasedly optimize using cross-validation. Second, cross-validation is needed to tune PRS p-value thresholds. Finally, cross-validation is needed for our new PRS pleiotropy metric. (Concretely, we estimate it would take >1 year for 100 jobs run in parallel to perform 10-fold cross-validated GWAS on our 217 phenotype matrix.)

iii) We fully acknowledge that GWAS summary statistics are available for most of the 217 phenotypes we use in UKB (in the white British individuals). However, this is currently the exception rather than the rule, especially for newer datasets or those with diverse ancestries. This is further exacerbated by the fact that optimizing inputs to MTAG requires evaluating summary statistics for many secondary traits, not just the focal trait. For example, summary statistics are not yet readily available for any of our other cohorts (UKB individuals with non-white British ancestries, ATLAS, or iPSYCH).

c. The second reason for not using 216 phenotypes in MTAG “MTAG accrues false positives as the number of traits grows” makes sense. In fact, so much so that one wonders if including 216 traits in the trait imputation may introduce more noise than if the number of traits were limited. It would be useful to benchmark MTAG.All by imputation that uses the same traits (or same traits + sex + age plus demographic traits).

This is a great idea to benchmark the different approaches. We added this experiment to the Main text:

*Finally, to compare like-to-like, we repeated our evaluation of SoftImpute imputation accuracy after restricting to the MTAG.All input phenotypes (and sex, age, and 20 PCs). We found that imputation performed much worse with this reduced set of phenotypes (**Supplementary Figure 2**). For LifetimeMDD, specifically, imputation R^2 dropped from 59.6% to 39.5% ($P < 2 \times 10^{-5}$, pooled t-test across folds). Overall, average imputation R^2 on MTAG.All traits dropped from 34.8% to 20.3% (**Supplementary Table 6**).*

(We interpret ‘demographic traits’ as PCs; broader definitions would include ~all 217 phenotypes.)

Intuitively, imputation is relatively robust to low-quality phenotypes because they have little impact on the imputed values, which is all that matters for downstream GWAS (Dahl 2016 Nat Genet). MTAG, on the other hand, jointly models multiple phenotypes, which becomes more statistically challenging and miscalibrated as #traits grows (Figure 1 in Turley 2018 Nat Genet).

Supplementary Figure 2. Comparison of our baseline imputation accuracy with softImpute to alternative approaches using different input phenotype matrices. Accuracy is compared to approaches that impute a phenotype matrix including: (A) only the MTAG.All traits plus demographic traits (age, sex, and 20 PCs); (B) only males; (C) only females; (D) adding a column for BMI. (E-H) show the p-values comparing the baseline R2 with the relevant alternative imputation approach. p-values are calculated based on pooled t-tests across 10 replicates of copy-masked data with 1% added missingness (Methods) for E-G; H uses paired t-tests instead because the copy-masks are identical in both imputation approaches.

3) Sex is used as a variable in the imputation, but it seems likely that the imputation could be improved by conducting it within sex?

This is a terrific suggestion. We have added this experiment as a new paragraph in our simulation results and display the results in Supplementary Figure 2:

As sex/gender significantly impacts MDD risk²⁵⁻²⁷, we repeated our SoftImpute analyses in each sex separately and compared the results to our primary approach (which includes sex as a column in the phenotype matrix). Overall, we find that the imputation R¹ are highly correlated across the female-, male-, and joint-imputation approaches (Pearson r between R¹female and

R¹joint = 0.928; between R¹male and R¹joint = 0.927; between R¹male and R¹female = 0.85,

Supplementary Figure 2), without any statistically significant differences on average (average

*R¹female=21.7%, R¹male=20.1%, R¹joint=21.3%, **Supplementary Table 2**).*

The average performances across phenotypes were not statistically significant (average R²_{female}=21.7%, R²_{male}=20.1%, R²_{joint}=21.3%). A few phenotypes had significantly different R² in the sex-specific models, roughly balanced between better (7 for males, 8 for females) and worse (12 for males, 11 for females).

The imputation accuracy for our focal phenotype LifetimeMDD is slightly worse in the sex-specific models ($R^2_{\text{female}}=38.6\%$, $R^2_{\text{male}}=34.3\%$, $R^2_{\text{joint}}=39.5\%$).

4) It seems strange that BMI was not included as a variable in the imputation. Also strange not considered in the pleiotropy analyses?

i) We repeated the imputation accuracy estimation after including BMI as an additional phenotype. The estimated imputation accuracies were almost identical ($R^2=99.9\%$). This is expected as no single variable has too great an impact on phenotype imputation. We include these results in Supplementary Figure 2, and describe the analysis in the figure's caption.

ii) We did not initially include BMI in the pleiotropy analyses (Figure 6) because we used the same phenotypes as in the phenotype imputation matrix (except genetic PCs and genotyping array). Due to the cost of running 10-fold cross-validated GWAS and the negligible impact of BMI on imputed LifetimeMDD, we chose not to add BMI as an additional point on Figure 6 (which includes >200 phenotypes).

5) Ideally the PGC29 cohorts could have been considered in Figure 5B as they are LifetimeMDD, but that is likely a big job. The UKB GWAS sumstats from the phenotype imputation and MTAG need to be made available to allow other to do this.

i) PGC29: We agree this would be great but a large amount of additional work. We discussed this with the chair of the MDD Working Group of the PGC Professor Cathryn Lewis and determined that this would be infeasible. We feel that our results from two iPSYCH cohorts, ATLAS, and non-British European ancestry individuals in UKB give sufficient evidence to substantiate our conclusions. We also feel that our analyses in UKB of PRS built from GWAS on iPSYCH, PGC29, and 23andMe add important context for our results.

ii) Thank you for catching that we should share summary statistics. We have made them available and added the link to the Data availability section: DOI: 10.6084/m9.figshare.19604335. We have also made all of our analysis code available at <https://github.com/andywdahl/mdd-impute> and <https://github.com/caina89/MDDImpute>.

Minor comments:

1. The term "effective" sample size usually refers to a specific power-related statement. For example, the effective sample size of a case control cohort is the power of the equivalent power of the actual cohort $N_{\text{case}} < N_{\text{control}}$ expressed as the power of a sample size where $N_{\text{case}} = N_{\text{control}}$. Here, although the power is certainly increased as a reflection of sample size, and the power doesn't reflect the actual sample size because essentially a correlated trait is measured, but the power is not quantified. So the use of "effective" should be avoided. Or is this the effective N measured from MTAG? In which case effective it is OK.

Thank you for pointing this out. There are three relevant definitions of effective sample size, which we

did not clearly delineate in our initial submission. All are power-related statements.

i) We were using the term as in genotype/phenotype imputation, meaning the power-equivalent number of observed genotypes/phenotypes (Marchini and Howie 2010 Nat Rev Genet, Dahl 2016 Nat Genet); Hormozdiari 2016 AJHG explicitly proves this in a special case of phenotype imputation (Eq 15) Concretely, we calculate this as $N_{obs} + N_{miss} * R^2$. We now (a) explicitly reference these papers in the main text when discussing effective sample size and (b) explicitly define effective sample size in the Methods section to eliminate the prior ambiguity:

*We define the effective sample size post-imputation as $N_{obs} + N_{miss} * R^2$, analogous to genotype imputation^{16,17,24}; we note that this approximates the power-equivalent number of observed phenotypes.*

This is a different definition of effective sample size from that in GWAS which accounts for the imbalance between number of cases and controls.

iii) We have added the definition about case/control imbalance to the Methods when describing different cohorts. We also emphasize the difference with the imputation-related definition:

*For all GWAS, we have indicated their effective sample sizes accounting for imbalance between cases and controls ($N_{eff} = 4/(1/N_{cases} + 1/N_{controls})$) in **Supplementary Tables 5 and 6**. We note this is different from the imputation-related definition of effective sample size, and also differs from MTAG's definition.*

iii) MTAG uses a third notion of effective sample size based on the power-equivalent sample sizes for MTAG GWAS vs single-trait GWAS. We show this in Figure 4 as one quantification of the power for each set of MTAG inputs. We have added this definition to the caption for Figure 4 to improve clarity:

*The MTAG effective sample size refers to the power-equivalent sample sizes of MTAG GWAS vs single-trait GWAS¹⁸, calculated as $N_{eff} = N_{single} * (\chi^2_{MTAG} - 1)/(\chi^2_{single} - 1)$, where χ^2 are the average GWAS chi-squared values.*

2. Suggest that the PRS in non-EUR is placed in supplement as it seems not very interpretable, but good to have tried.

We agree that these results are challenging to interpret, and perhaps even weaken the appearance of our approach. Indeed, we did put several panels that were less interpretable into Supplementary Figure 9. Nonetheless, we feel it is important to keep our results in CONVERGE and African ancestry individuals in UKB in Figure 5 to substantiate our Discussion points on the importance of the future work needed to improve portability:

On the other hand, we found that phenotype integration can improve PRS portability across ancestries. However, the results are less clear than those in European ancestries, highlighting the need for greater sample sizes in diverse ancestries and methods for cross-ancestry portability.

3. This section “Phenotype integration improves PRS portability” should be “Phenotype integration and MTAG improve PRS portability”.

i) As noted above, we define phenotype integration as the union of phenotype imputation and MTAG.

ii) To improve clarity, we have changed this section title to “Phenotype imputation and MTAG improve PRS portability”. We have also made this change to all other section titles with “phenotype integration”.

4. SNP-based heritability is now the preferred term as SNP-heritability is ambiguous.

Good point. We have made the suggested change.

5. When introducing the cohorts 23andMe, PGC29 etc in the written it would be good to add effective N (as in usual definition!) as a quick benchmark.

Thank you, we have now done this. (Please also see our answer to Minor #1 above)

6. More detail is needed for the estimates of SNP-based heritability. Make a table with N is used in LD score regression and the K factor for conversion to liability and outcomes. These data will help other researchers accessing the sumstats.

We agree this is helpful and have made a new Supplementary Table 5 containing all this information.

7. Fig 6. Use wording in legend to make it clear that the LifetimeMDD and GPsy are the same in all figures.

We agree this is confusing, and have added the following clarification to the caption in Figure 6:

Note that GPsy and LifetimeMDD are each used in two ways: To build the PRS, and to evaluate PRS Pleiotropy.

8. Extended Data Fig 1. Since LifetimeMDD is the key trait, can you add a 2x2 table of Measured/predicted data that generates the brown dot in part a?

Softimpute (and AutoComplete) do not impute binary case/control labels, but rather continuous variables (akin to latent liabilities). Thus we have compared the imputed values for held out cases vs

controls by comparing their distributions (like a 2xInfinity table). We think this is an informative figure and have added it as Panel D in Extended Data Figure 1, and we describe it in the main text by:

We show the distribution of imputed LifetimeMDD values in held-out cases and controls in Extended Data Figure 1.

If we threshold these imputed values at 0.5, we get the following 2x2 confusion matrix. These numbers correspond to held-out observations of LifetimeMDD (combined across 10 replicate experiments that each hold out ~1% of LifetimeMDD observations).

	Obs Ctrl	Obs
Case Imp Ctrl	611	2202
Imp Case	10	121

Reviewer #2:

Remarks to the Author:

This is an interesting study that uses phenotype imputation to derive MDD-like traits that are similar in their associations to a CIDI-derived MDD diagnosis. The findings could lead to larger sample sizes from UK Biobank and other studies and might lead to results that are more relevant to MDD research, than the broad phenotypes currently widely-used.

We thank the reviewer for their insights into missing data imputation and depression phenotyping. Our hope is indeed that our work will produce an improvement in the quality of the “shallow” phenotyping often being used for work on MD in the UKB and elsewhere, which we feel is at risk of misleading the field. Responding to these comments has significantly expanded the way we reference related work, which we think adds important context, and has added important caution to readers regarding potential pitfalls with imputation.

The work is somewhat novel in MDD, although there are other examples of phenotype imputation, such as <https://www.biorxiv.org/content/10.1101/2022.08.15.503991v1.abstract> (using a more complex approach) and Glanville (<https://www.cambridge.org/core/journals/bjpsych-open/article/multiple-measures-of-depression-to-enhance-validity-of-major-depressive-disorder-in-the-uk/D34B7DCDD744B0C3520C7AEAE6A8E718>). The results aren't entirely novel, but takes a different methodological approach to these studies.

i) We agree that phenotype imputation is not novel. We wrote a paper on it >5 years ago (Dahl 2016 Nat Genet), and our current paper uses a method from 2010 (softImpute). However, phenotype imputation is rarely used in large-scale GWAS. We feel the main barrier to adoption is the limited evidence that imputed phenotypes yield disorder-specific results in complex human disorders. Here, we develop a framework to answer this question and we deploy it in the context of UKB-based MDD GWAS, a sample much examined in the research literature..

ii) The cited Autocomplete paper is a companion to our paper (we wrote them together). We study Autocomplete extensively in our paper. We appreciate the enthusiasm about this method, and we agree that it is important work.

iii) In brief, Glanville et al constructs several new MDD measures by manually combining observed shallow depression measures. This is highly relevant, and is analogous to our prior work constructing LifetimeMDD (Cai 2020 Nat Genet) and work that constructs phenotypes based on proxies of depression (Okbay 2016 Nat Genet) or family history (Liu 2017 Nat Genet). However, these efforts are not phenotype imputation. Please see our full response in response to your last comment.

The authors state that MDD's treatments are 'relatively ineffective' but make no comparison. MDD treatments compare well in terms of NNT to many for physical disorders and are similarly effective to many psychological and physical treatments for other psychiatric disorders. I think this statement is oversimplified and misleading and should either be made more explicit (with % treatment failures), compared to treatments with other disorders, or removed.

We appreciate this insight and have removed this comment. We do not wish to distract from our main points.

The shallow-deep phenotype divide is framed in a way that can't currently be confirmed or supported using the studies they cite. They state that deeply-phenotyped studies have produced smaller numbers of loci that could generate hypotheses on MDD biology. However, that does not stand up to scrutiny – the studies they cite are volunteer studies and volunteers in studies of psychiatric disorders are frequently better educated and resourced than the population at large. Biological inferences from these studies are highly problematic when there are very few (sometimes zero) replicated loci – as those inferences also depend on the number of significant loci and genetic signals that can be partitioned into pathways and processes (for example).

We don't quite understand the overall thrust of this comment. But we do see several interesting and independent points about deep and shallow MDD GWAS, which we respond to separately:

i) "Biological inferences from these studies are highly problematic when there are very few (sometimes zero) replicated loci"

We strongly agree that replication is essential for validating GWAS results, generally, and phenotype integration, specifically. Our new GWAS hits replicate well in several cohorts:

We found that all 8 hits shared between both ImpOnly GWAS have sign-consistent effect size estimates across all of these depression cohorts, as well as neuroticism. Moreover, all 8 are significant for observed LifetimeMDD in UKB at $P < 0.05/23$. Finally, out of the 23 SNPs significant in one ImpOnly GWAS, 18 replicate in at least one GWAS of observed MDD at $P < 0.05/23$ (Extended Data Figure 2).

ii) "the studies they cite are volunteer studies"

We completely agree this is worrisome in MDD GWAS, especially for MHQ-derived phenotypes. This limitation is prominent in our Discussion, and we have now significantly expanded it:

We have worked on the deepest MDD phenotype in UKB, LifetimeMDD, which is derived by applying DSM-5 criteria in silico to self-rated MDD symptoms in the MHQ. This is in fact shallow compared to a clinical diagnosis based on a structured in-person interview, especially due to self-report biases and misdiagnoses that have been repeatedly demonstrated⁶¹⁻⁶⁴. In the future, this bias could be mitigated using methods based on probability weights, which have been recently developed for GWAS applications^{65,66}.

Health record measures of MDD are also grouped with other broad phenotypes, although these are perhaps the most representative of the population as a whole and most likely to resemble those who would gain from insights and treatments developed on the back of GWAS. The iPSYCH study mentioned is also one whose methods are based partly on Danish electronic health records.

We understand the reviewer's concern with conceptually lumping together health records with broad/shallow/self-reported phenotypes. We completely agree that ascertainment and public health relevance of EHR-based MDD are likely quite different from self-reported MDD.

i) We corrected our nomenclature in the Introduction, which previously grouped together self-reported and EHR-based MDD phenotypes:

This motivates the use of shallow phenotypes in large biobanks, including self-reported depression or depression treatment^{3,5}. Sample sizes are often further increased by using health records of seeking care

for depression (e.g., iPSYCH¹⁴, Million Veterans Project⁶). This more accurately represents the population because it is not based on volunteering, though the accuracy and consistency of diagnostic criteria will vary among studies and study sites.

ii) We strongly agree that it is important to validate our findings in datasets in EHR-based data. As such, we validated our improved PRS performance from phenotype integration using two EHR-based cohorts (ATLAS and iPSYCH)

iii) It is not clear whether EHR codes or CIDI-based LifetimeMDD (based on self-report) is better in terms of representativeness. EHR codes do not require volunteering (although they do require people to go to a doctor and receive ICD codes, which has other ascertainment bias).

iv) Health records are not always high quality. These range from ICD codes (who could be keyed by any professional) to detailed electronic records from clinical interview studies based on clinical samples. While the latter are as close to a “gold-standard” as we have in the field, the former are highly liable to false positives (Mitchell 2009 Lancet).

v) Overall, in terms of ‘likelihood to benefit from treatments and insights’, we previously demonstrated that LifetimeMDD captures more MDD-specific signals, and therefore argue it is superior for investigating the biology and treatments of MDD (Cai 2020 Nat Genet).

The statement that broad phenotypes generate signals that are not specific for MDD is also questionable when those samples lack power, and an equal comparison (without down-sampling broad phenotyping studies) is problematic as the ground truth is not known.

We think this point is about teasing apart the contributions to power from sample size vs phenotype depth. We previously compared GWAS based on deep vs down-sampled shallow (aka broad) phenotypes (Cai 2020 Nat Genet). We found that the deeper phenotypes yielded better power when sample sizes were matched, yet that shallow phenotypes had more power with the real sample sizes.

This manuscript essentially reports missing phenotype imputation. There is an enormous literature on this topic, and discussion of the pitfalls and limitations. Reference to this literature is largely missing from the paper.

Please see our comments above regarding the novelty of phenotype imputation.

The CIDI-SF is assumed to be the preferred phenotype in UKB. People who completed this questionnaire are atypical of the whole in several respects (Adams M, 2018 IJE: <https://academic.oup.com/ije/article/49/2/410/5526901>). On the back of UKB, whose participation rate was ~6%, this sample is clearly very highly selected. Have the authors considered the biases that may result from this selection?

- i) Our preferred phenotype in UKB is LifetimeMDD, which is derived from applying DSM-V criteria to the CIDI-SF. We previously constructed this phenotype and showed that it captures more MDD- specific genetic effects than shallower/broader measures, e.g. ICD10-code based measures, or self- reported depression (Cai 2020 Nat Genet).
- ii) We strongly agree that ascertainment bias in UKB and the MHQ is an important limitation. We cited this paper in the Discussion section on key limitations of our study. We have now further emphasized this point and suggested potential solutions for future work with a new sentence:

This is in fact shallow compared to a clinical diagnosis based on a structured in-person interview, especially due to self-report biases and misdiagnoses that have been repeatedly demonstrated^{61–64}. In the future, this bias could be mitigated using methods based on probability weights, which have been recently developed for GWAS applications^{65,66}.

- iii) Our external validation in ATLAS and iPSYCH (Figure 5) shows that our primary results hold for non-volunteer MDD phenotypes. This is especially true for iPSYCH, which contains all diagnosed cases in Denmark.
- iv) While important, we note that this criticism applies to all prior studies of the UKB MHQ data, and the more general point about UKB ascertainment also applies to all prior studies of UKB and most genetic studies.

One of the many limitations of missing value imputation is that it reduces the variance of the phenotype of interest which, if observed, would deviate randomly from the proposed model used to derive missing values. Can the authors clarify whether this was something they saw in the imputed data and whether they think it's an important consideration?

We agree the imputed data have lower variance, and that single imputation approaches can be miscalibrated (the textbook solution is multiple imputation, as the reviewer likely knows, but this is not straightforward for complex imputation methods or high-dimensional data). But all models are wrong—our question is whether our approach is useful for biobank-based GWAS. We use extensive genetic tests and external data to show that single imputation (and, importantly, MTAG) can work well for MDD in UKB.

Nonetheless, we agree that the limitations of single imputation are important and relevant, and moreover that we did not sufficiently emphasize them. We have now added new analyses to characterize the extent of these biases in the moments of the imputed phenotypes (below), and we have added text to emphasize this important caveat to the Main text:

We found that the imputed measures had deflated variances and inflated correlations (Supplementary Methods, Supplementary Figure 1), as expected¹⁶. This could bias some

downstream tests, such as genetic correlations. One main goal in this work is to determine if this approach to phenotype imputation succeeds for large scale single-trait genetic studies.

We have also added this caveat when discussing the combination of MTAG and imputation:

Finally, we could use GWAS on imputed phenotypes as inputs for MTAG; however, this may exacerbate biases in imputation because MTAG leverages correlations between traits, which are biased by most imputation approaches (Supplementary Figure 1)

In greater detail:

i) We plotted the variance of imputed vs observed phenotypes (Supplementary Figure 1A). The variances were deflated by imputation, as expected. The average deflation in standard deviation was 20.3% (10th percentile=8.2%, 90th percentile=36.0%).

iii) We plotted the correlation within imputed phenotypes vs within observed phenotypes (Supplementary Figure 1B). The correlations were inflated by imputation, as expected. The relative increase in correlation between LifetimeMDD and other traits had median 3.95 (10th percentile=1.35, 90th percentile=13.3). (Note: the imputed covariances are actually smaller than the observed covariances because their variances are smaller.)

iii) In principle, heteroscedasticity could be accommodated with a weighted least squares. (We have argued elsewhere this is essential for other tests, e.g. for polygenic GxE (Dahl 2020 AJHG) or vQTLs (Musharoff 2018 bioRxiv).) We have now added a comment to the Discussion on this future direction. We have also added a classic textbook reference to multiple imputation (Little and Rubin) and a reference to a very recent preprint using a related idea to correct imputation downstream (McCaw 2022 bioRxiv):

*One limitation of our specific imputation approaches is that they distort higher-order moments (**Supplementary Figure 1**), which will bias some downstream analyses, like genetic correlation. As such, it is essential to thoroughly validate results with external data. In future work, this could be addressed with multiple imputation⁵³ or with downstream tests that allow different effect sizes or noise variances between imputed and observed phenotypes⁵⁴.*

The MDD-related traits used for imputation are the ones correlated with MDD in the training sample. These are presumably considered to be those that are relevant by the authors, but it's clear that in the imputed samples those variables will also be correlated with MDD, maybe more so than if they had been measured directly. Do the authors think this is a problem? Do the authors think that their analyses could be accused of being circular – that pattern of genetic correlations used to support their methods are influenced by the traits used for imputation?

“in the imputed samples those variables will also be correlated with MDD, maybe more so than if they had been measured directly”

This is a great point. Initially, we addressed this only in a limited way by comparing genetic

correlations of observed/imputed measures to external GWAS on MDD and neuroticism (Figures 2 and 4) to demonstrate that imputed LifetimeMDD is highly genetically correlated with observed MDD. We have now also added Supplementary Figure 2 which compares the phenotypic correlations between observed vs imputed phenotypes (please see our above point).

“Do the authors think that their analyses could be accused of being circular – that pattern of genetic correlations used to support their methods are influenced by the traits used for imputation?”

i) Our approach is not circular. The definitive empirical proof is that phenotype integration improves PRS accuracy in external data (iPSYCH, ATLAS, non-white British UKB individuals). Circular reasoning in the training data could not possibly improve power in external data.

ii) Nonetheless, we do agree there is some circularity in using correlations to validate imputation. However, there are three reasons why we don't consider this problematic. Most importantly, we view these analyses as sanity checks and illustrations for less-familiar readers (comparing correlations is a very small part of our paper). Second, we compared genetic correlations, which is less circular/trivial than ordinary phenotypic correlations (though we acknowledge that phenotypic and genetic correlations are often very similar). Finally, it is not entirely trivial that imputed correlations resemble imputed correlations—indeed, we are happy to have added a systematic exploration of these correlations in Supplementary Figure 2 in response to an above concern that these correlations may be too different.

The increased number of GWAS loci identified in the observed vs imputed MDD GWAS analysis is not a good indication of power when the number of gwsig loci is low. Repeated GWAS analyses of the same phenotype in samples of equal size can generate different numbers of loci by the play of chance alone. Do the authors think there is a better approach to this question? If they took effect sizes from another independent GWAS-MA study, would that help to identify whether they had more power to discover significant signals using a fairer test?

i) “The increased number of GWAS loci identified in the observed vs imputed MDD GWAS analysis is not a good indication of power when the number of gwsig loci is low. Repeated GWAS analyses of the same phenotype in samples of equal size can generate different numbers of loci by the play of chance alone.”

Because the differences were so large (e.g., 26/40/33 for softImpute/autocomplete/MTAG.All vs 1 for observed LifetimeMDD), our intuition was that they must be significant. However, we strongly agree this important point must be formally tested.

We have now added a test for significant differences in the number of hits across our imputed and observed GWAS. Specifically, we modeled the number of hits in each GWAS j as independent* binomial random variables, where each of N truly causal** loci is a GWAS hit with probability p_j . We then test whether $p_j = p_{j'}$ for each pair of GWAS j and j' .

We confirmed that these improvements in the number of GWAS hits over the single hit from observed LifetimeMDD are very unlikely to result purely from chance (**Supplementary Figure 3**).

Overall, we appreciate this suggestion and think this addition is important because it shows unequivocally that phenotype integration does add power.

*this is conservative because we expect these binomial variables to have positive dependence, because they are each GWAS of correlated traits in a single population

**this assume equal power at each causal locus, which is not exactly correct but is plausibly accurate for complex traits

Supplementary Figure 3. Binomial tests for equal power between each pair of 5 MDD GWAS in UKB on unrelated white British individuals. Each point on a curve tests the hypothesis that GWAS 1 and GWAS 2 have equal power to detect each of N causal loci. Specifically, for each number of causal loci N, we assume that the number of GWAS hits for each GWAS is independently distributed $n_i \sim \text{binomial}(N, p_i)$ and then test whether $n_1 = n_2$ using Pearson's chi-squared test. Our test shows that GPpsy and phenotype integration (imputed or MTAG.All) GWAS are all more powerful than the (observed) LifetimeMDD GWAS; except for implausibly few causal loci (e.g., $N < 100$), our test does not detect different differences between phenotype integration GWAS.

“Do the authors think there is a better approach to this question? If they took effect sizes from another independent GWAS-MA study, would that help to identify whether they had more power to discover significant signals using a fairer test?”

i) We think that our above test is fair and proves the number of added GWAS hits is significant.

ii) We did take effect sizes from multiple other studies to determine whether the signals we discovered were valid. We used a very strict replication test by evaluating only the GWAS hits that were found in GWAS of ImpOnly but not in GWAS of observed LifetimeMDD. We found that most of these 23 novel associations replicated in at least one external cohort (18/23, at $p < 0.05/23$) and only a small proportion of the GWAS hits from ImpOnly GWASes (4/13 and 6/18 respectively for SoftImpute and Autocomplete) showed significant effect heterogeneity (at $p < 0.05/23$) from that in observed LifetimeMDD (Extended Data Figure 2). This strongly suggests that our approach adds significant GWAS power, which is a helpful complement to the formal significance test above.

Why were heritability estimates on p5 provided on the observed rather than on the liability scale?

We now have changed all SNP-based heritability estimates to the liability scale in the main text and Figure 2F. We have kept both scales in Figure 4D. We have also added Supplementary Table 4 with further details, such as the assumed prevalence we used to convert between scales (using Lee 2011 AJHG).

The authors quote a paper by Glanville et al that used a different approach to missing values – creating a composite phenotype that also enlarged sample size. There aren't any direct comparisons with this approach. I think those would be interesting and useful.

We thank the reviewer for emphasizing this relevant paper. We think Glanville 2021 BJPsych Open takes a very different approach from our own.

First, it is not phenotype integration. Rather, it constructs a new phenotype which can be used as a proxy for phenotyped CIDI-based MDD. This is directly comparable to our prior work constructing LifetimeMDD (Cai 2020 Nat Genet), as was discussed in Glanville et al. Conceptually, it is very similar to GWAS by proxy methods proposed previously (GWAX, Liu et 2017 Nat Genet).

Second, Glanville et al never performed GWAS. As such, we cannot evaluate this proposed phenotype based on GWAS summary statistics, as we did with PGC, 23andMe or iPSYCH. Performing GWAS on the Glanville phenotypes (they construct several) would be a valuable extension to that work. This can be its own project, and is out of scope for our current paper. We therefore propose this future work in Discussion:

In the future, PRS Pleiotropy can be used to evaluate newly constructed phenotypes, including manual combinations of existing measures¹³.

Third, our MTAG.AllDep GWAS is intuitively closely related to the Glanville et al phenotypes because it uses similar input phenotypes. For the specific purpose of GWAS, MTAG is expected to be a near-optimal approach to combining these measures. We have now added this point when describing MTAG:

We note that MTAG.AllDep is analogous to depression phenotypes derived by manually combining similar input phenotypes¹³ and that MTAG.FamilyHistory is analogous to prior approaches that integrate family history measures into GWAS^{43,44}.

Reviewer #3:

Remarks to the Author:

This is a fantastic paper that is well written and explores important empirical and methodological questions. In this paper, the authors discuss the trade-off between power and specificity. For many phenotypes, researchers must choose between small samples that are well phenotyped and large samples with less precise measurement. Poorly measured phenotypes can lead to reduced power (at best) and systematic biases in GWAS results (at worst). Similarly, methods that claim to increase power (such as MTAG) may also induce these sorts of biases. The authors use the application of major depressive disorder (MDD) to assess how these concerns play out in that setting. The propose a series of imputation approaches that can boost power without large risks of contamination and loss of specificity.

We appreciate the pointers to highly-relevant work which we previously missed, which dovetail nicely with our work. We also appreciate the insightful substantive comments, which have led us to change our own thinking around some high-level interpretation issues and to address two key limitations of our PRS pleiotropy metric.

Overall, I think the paper is in great shape. I have a small number of recommendations that the authors may consider integrating into a revision to clarify some of their analyses and connect it to existing work.

1) A very similar question is explored theoretically in Okbay et al. (2016). (See Supplementary Note Section 3.) How does the authors' analysis compare to the work done in that paper?

This is a very relevant reference that we missed. We completely agree that Supplementary Note Section 2 ("Tradeoff between sample size and phenotype heterogeneity") is a clear theoretical description of exactly our question in the case where there is a single 'deep' phenotype and a single secondary 'shallow' phenotype (which we think was a first step toward the theory behind MTAG). This work is a great complement to ours. It is formally rigorous with explicit assumptions, while our work explores a real high-dimensional application. As expected, these two paths converge on the same story: Meta-analyses inevitably face power-specificity tradeoffs.

Because this theory is so well-aligned with our intuition and results, adding this theoretical backbone to our Discussion was seamless and significantly strengthens our paper:

This metric characterizes a power-specificity tradeoff for MTAG, where adding more phenotypes generally increases power but sacrifices specificity. This empirical measure complements prior formal theoretical derivations of power-specificity tradeoffs in meta-analysis of heterogeneous traits⁷⁴.

2) Hill et al. (2019) also noted that MTAG-based summary statistics have greater power but seem to be less specific. For example, the genetic correlation between intelligence and educational attainment is 0.7 when using the raw summary statistics, but it's >.8 when using MTAG-based summary statistics for intelligence that leverage the educational attainment summary statistics. While this paper is a much more thorough treatment of that issue, I believe that this is the same phenomenon that the authors of this paper find when they discuss specificity. Do the authors agree or is there something unique about MDD?

First, this is a helpful reference that we missed. In terms of genetic correlation, we agree this is entirely the same phenomenon. We have now added this reference when we discuss our analogous results:

Third, neuroticism is significantly more genetically correlated with MTAG.Envs ($r_G = 0.84$, $SE = 0.01$) than LifetimeMDD ($r_G = 0.66$, $SE = 0.06$). These results are consistent with prior observations that MTAG-based summary statistics modestly inflate genetic correlation to the input phenotypes⁴⁵. Overall, the genetic correlations between MTAG and ...

Second, while the impact of MTAG on downstream r_G estimates will depend on the input r_G and h^2 , the polygenicity in the MTAG GWAS will also depend heavily on the polygenicity of the input focal and secondary traits. This will be highly trait specific, and is particularly relevant to MDD given that many GWAS loci are established for shallow phenotypes yet few are established for deep phenotypes. This isn't a qualitative distinction, but we hope this helps explain why we think MDD is particularly interesting and challenging.

3) I am having a hard time interpreting the PRS pleiotropy measure proposed by the authors. First, if the baseline R^2 for LifetimeMDD is different for the different PRSs, then how comparable is the ratio across PRSs. Given that R^2 is nonlinear in the effective sample size of the underlying GWAS, would we expect the PRS pleiotropy to be the same if we increased the sample size of the deep MDD phenotypes such that the baseline R^2 was the same? Relatedly, the authors use PRSice to construct their PRSs in this paper. This approach only uses a subset of SNPs that meet some p-value threshold. As a result, GWASs with greater power will use more SNPs that have smaller associations with MDD and which may be more pleiotropic in general. How does the PRS Pleiotropy compare when using the same set of SNPs in the various PRSs but using the different sets of GWAS estimates considered?

We thank the reviewer for bringing up these points. We have performed a series of analyses to address them, which significantly strengthened our paper and our own understanding of this nascent metric. Overall, we find that PRS pleiotropy is robust to the different sample sizes considered in our study.

First, we directly investigated the effect of sample size by down-sampling. Specifically, we repeated our PRS pleiotropy analyses after down-sampling Soft-ImpAll (N=337,126) and GPpsy (N=332,629) to 50K and 100K. This involved 10-fold cross-validated GWAS and PRS construction, including tuning the optimal p-value threshold per each cross-validated PRS). We studied the same 62 phenotypes where all full-sampled PRS gave significant predictions (used in Figure 6)

We found that down-sampled PRS Pleiotropy are very highly correlated to full-sample PRS Pleiotropy (Soft-ImpAll: Pearson r for 50K = 0.98; Pearson r for 100K = 0.99; GPpsy: Pearson r for 50K = 0.95; Pearson r for 100K = 0.99). While GPpsy at N=50K had significantly higher mean PRS Pleiotropy (two-sided paired t-test $P = 3.16 \times 10^{-11}$, mean difference = 10.2%), this difference vanishes at N=100K ($P = 0.727$, mean difference = 0.14%). Down-sampled Soft-ImpAll does not show significant difference in mean PRS Pleiotropy at either 50K ($P = 0.90$, mean difference = -0.06%) or 100K ($P = 0.70$, mean difference = 0.10%).

We show in the figure below (also added as Extended Data Figure 5) the full and down-sampled PRS Pleiotropy across all 62 phenotypes ordered by their respective full-sample PRS Pleiotropy (panel B,D). Consistent with the above statistics, there were nontrivial fluctuations for N=50K yet surprisingly little fluctuation for N=100K. The variations were more extreme for GPpsy than Soft-ImpAll, which we attribute to differences in R^2 for LifetimeMDD (which is likely the factor through which sample size acts).

Overall, we conclude that PRS Pleiotropy is qualitatively stable for a single PRS as a function of sample size (because correlations are high) and that it can be compared across different PRS for $N > 100K$ (though we cannot evaluate $N > 300K$). In the context of our study, this is not problematic because only observed LifetimeMDD has $N < 100K$ ($N = 67K$), which only means that our baseline is conservative.

We then went further to understand how sample size affects our PRS. Specifically, we investigated how sample size affects optimal P value thresholds, the number of included SNPs, and the variability of these statistics across folds of the data (Supplementary Figures 10 and 11, copied below). In summary, we find that:

- The down-sampled PRS thresholds are more liberal (panel A) and more variable (panel B) and
- The down-sampled PRS include more SNPs (panel C), which shows the net effect of more liberal thresholds (increasing #SNPs) and lower power (decreasing #SNPs)
- Down-sampled PRS give more variable R^2 across folds (as measured by coefficient of

variation, panel E), which is expected but distinct from the fact that the R2 decrease

- This results in higher variance in PRS Pleiotropy when using down-sampled PRS (panel F, especially for N=50K), which is expected but distinct from the fact that the average PRS Pleiotropy increases (especially for N=50K)

Supplementary Figure 10 (GPpsy):

Supplementary Figure 11 (Soft-impAll):

We think these investigations are extremely helpful in characterizing our PRS Pleiotropy metric, and Supplementary Figures 10-11 also have independent value for empirically characterizing the behavior of PRS. We have added a description of these analyses and key results to the Supplementary Note, which we briefly describe and reference in the Main text by:

Because of the complex relationship between effective sample size, p-value threshold, and PRS R2, we evaluated PRS Pleiotropy after downsampling the GWAS used to build the PRS (Supplementary Note). Overall, we found that PRS Pleiotropy is stable, though it can be upwardly biased for sample sizes below 100K (which is likely due to low power). In particular, our results are robust to the slight differences in the

training PRS sample sizes, though observed LifetimeMDD PRS Pleiotropy is a conservative baseline because it is trained on 67K individuals (Extended Data Figure 5, Supplementary Figures 10-11).

Finally, we answer the specific question: "How does the PRS Pleiotropy compare when using the same set of SNPs in the various PRSs but using the different sets of GWAS estimates considered?"

We considered two sets of SNPs: (A) 136,563 LD-pruned SNPs ($r^2 < 0.2$) in UKB, and (B) 91,315 SNPs from PGC29 clumped at $P_{\text{threshold}} = 1$). We find that the results we observed in our primary analyses (Figure 6) persist with both sets of SNPs. We have added these results as Extended Data Figure 6 (copied below). We have fully described this analysis in the Supplementary Note, and added a short description to the Main text:

Further, we confirmed that these results persist when we use exactly the same SNPs in each PRS (Supplementary Note, Extended Data Figure 6).

4) On page 8, the authors claim "If phenotype integration captures core MDD biology, it will improve PRS predictions regardless of the context". This also seems a bit too strong. Why can't core biology operate

through context dependent pathways? For example, the main SNP on CHRNA5 that predicts smoking behavior will operate more strongly in communities where smoking is more prevalent. You could imagine genes that play a core biological function by affecting the way the body responds to environmental factors that vary by context. The only way I can think that the above statement is necessarily true is if "core MDD biology" is defined as factors that are independent of context, in which case the statement is somewhat tautological.

We agree. This was an overstatement and we appreciate you catching it. We made two mistakes: (1) lumping together context specificity (and/or stratification bias) with 'core vs peripheral' biology, and also (2) making a basic logic error.

Re: (1), The point we hoped to get across is that one could imagine population stratification (or some other context specific to white British individuals in UKB) drives our increased genetic signals (GWAS hits and PRS R2). This risk is mitigated by comparing across ancestries (and other contexts). But this is not directly related to 'core vs peripheral' biology, which is not relevant and we should not have mentioned.

Re: (2), our inted point is that portability → 'more likely causal', which we stand by (and is the second half of this sentence). However, we stated the converse in the first half of the sentence (casual → definitely portable), which we agree is not generally true

We have fixed our incorrect statement by centering on dataset-specific bias (removing 'core') and fixing the logic error (removing first half of the sentence):

Demonstrating portability is essential in order to establish that phenotype integration does not merely reflect dataset-specific biases.

As an aside, we still support this converse statement as a rough hypothesis – the more direct the SNP to phenotype pathway, the more likely the effect is to port between contexts. But, again, we acknowledge this was overstated as a known and general fact rather than a rough hypothesis.

5) Typo: I think there may be a missing word on page 9, in section titled "Phenotype integration improves PRS portability." It currently says, "Finally, we also for PRS prediction accuracy in UKB individuals with Asian ancestry..."

This is helpful, thank you. We have fixed this now.

References:

Okbay, A., Baselmans, B. M., De Neve, J. E., Turley, P., Nivard, M. G., Fontana, M. A., ... & Rich, S. S. (2016). Genetic variants associated with subjective well-being, depressive symptoms, and

neuroticism identified through genome-wide analyses. *Nature genetics*, 48(6), 624-633.

Hill, W. D., Marioni, R. E., Maghzian, O., Ritchie, S. J., Hagenaars, S. P., McIntosh, A. M., ... & Deary, I. J. (2019). A combined analysis of genetically correlated traits identifies 187 loci and a role for neurogenesis and myelination in intelligence. *Molecular psychiatry*, 24(2), 169-181.

Decision Letter, first revision:

7th February 2023

Dear Na,

Your revised manuscript "Phenotype integration improves power and preserves specificity in biobank-based genetic studies of MDD" (NG-A60706R) has been seen by the original referees. As you will see from their comments below, they find that the paper has improved in revision, and therefore we will be happy in principle to publish it in *Nature Genetics* as an Article pending final revisions to comply with our editorial and formatting guidelines.

We are now performing detailed checks on your paper, and we will send you a checklist detailing our editorial and formatting requirements soon. Please do not upload the final materials or make any revisions until you receive this additional information from us.

Thank you again for your interest in *Nature Genetics*. Please do not hesitate to contact me if you have any questions.

Sincerely,
Kyle

Kyle Vogan, PhD
Senior Editor
Nature Genetics
<https://orcid.org/0000-0001-9565-9665>

Reviewer #1 (Remarks to the Author):

The authors have addressed the comments I raised

Reviewer #2 (Remarks to the Author):

I have no further comments on the manuscript

Reviewer #3 (Remarks to the Author):

Thanks to the authors for a comprehensive response to each of my questions. I have no further concerns.

Final Decision Letter:

18th September 2023

Dear Na,

I am delighted to say that your manuscript "Phenotype integration improves power and preserves specificity in biobank-based genetic studies of major depressive disorder" has been accepted for publication in an upcoming issue of Nature Genetics.

Your paper will be published online after we receive your corrections and will appear in print in the next available issue. You can find out your date of online publication by contacting the Nature Press Office (press@nature.com) after sending your e-proof corrections. Now is the time to inform your Public Relations or Press Office about your paper, as they might be interested in promoting its publication. This will allow them time to prepare an accurate and satisfactory press release. Include your manuscript tracking number (NG-A60706R1) and the name of the journal, which they will need when they contact our Press Office.

Before your paper is published online, we will be distributing a press release to news organizations worldwide, which may very well include details of your work. We are happy for your institution or funding agency to prepare its own press release, but it must mention the embargo date and Nature Genetics. Our Press Office may contact you closer to the time of publication, but if you or your Press Office have any enquiries in the meantime, please contact press@nature.com.

Acceptance is conditional on the data in the manuscript not being published elsewhere, or announced

in the print or electronic media, until the embargo/publication date. These restrictions are not intended to deter you from presenting your data at academic meetings and conferences, but any enquiries from the media about papers not yet scheduled for publication should be referred to us.

Please note that Nature Genetics is a Transformative Journal (TJ). Authors may publish their research with us through the traditional subscription access route or make their paper immediately open access through payment of an article-processing charge (APC). Authors will not be required to make a final decision about access to their article until it has been accepted. [Find out more about Transformative Journals](https://www.springernature.com/gp/open-research/transformative-journals)

Authors may need to take specific actions to achieve [compliance](https://www.springernature.com/gp/open-research/funding/policy-compliance-faqs) with funder and institutional open access mandates. If your research is supported by a funder that requires immediate open access (e.g. according to [Plan S principles](https://www.springernature.com/gp/open-research/plan-s-compliance)), then you should select the gold OA route, and we will direct you to the compliant route where possible. For authors selecting the subscription publication route, the journal's standard licensing terms will need to be accepted, including [self-archiving-and-license-to-publish](https://www.nature.com/nature-portfolio/editorial-policies/self-archiving-and-license-to-publish). Those licensing terms will supersede any other terms that the author or any third party may assert apply to any version of the manuscript.

If you have not already done so, we invite you to upload the step-by-step protocols used in this manuscript to the Protocols Exchange, part of our on-line web resource, natureprotocols.com. If you

complete the upload by the time you receive your manuscript proofs, we can insert links in your article that lead directly to the protocol details. Your protocol will be made freely available upon publication of your paper. By participating in natureprotocols.com, you are enabling researchers to more readily reproduce or adapt the methodology you use. Natureprotocols.com is fully searchable, providing your protocols and paper with increased utility and visibility. Please submit your protocol to <https://protocolexchange.researchsquare.com/>. After entering your nature.com username and password you will need to enter your manuscript number (NG-A60706R1). Further information can be found at <https://www.nature.com/nature-portfolio/editorial-policies/reporting-standards#protocols>

Sincerely,
Kyle

Kyle Vogan, PhD
Senior Editor
Nature Genetics
<https://orcid.org/0000-0001-9565-9665>